# Structural order differentiation unlocks the energy storage performance of commensurate antiferroelectric ceramics

Guanglong Ge[1,5], Jin Qian[1,5], Cheng Shi[1], Chao Sun[1], Simin Wang[1], Hongguang Wang [2] ✉, Tengfei Hu [3,4] ✉, Peter A. van Aken [2], Bo Shen[1] & Jiwei Zhai [1] ✉

Commensurate modulated antiferroelectric ceramics exhibit limited application prospects, a quasi transient antiferroelectric-ferroelectric phase transition has locked their energy storage performance. Highly homogeneous oxygen octahedra set produce only one type of antiferrodistortion-ferrodistortion transition, followed by a rapid triggering of the antiferroelectric-ferroelectric phase transition. Here, we propose a strategy of structural order differentiation engineering to disrupt the homogeneity of oxygen octahedra by initiator/enhancer co-substitution, and we have successfully unlocked the energy storage performance of commensurate modulated antiferroelectric ceramics. By constructing oxygen octahedra sets with highly differentiated rotational distortions, an energy storage density of 23.11 J/cm$^3$, an energy storage efficiency of 85.55%, and a discharge energy density of up to 16.45 J/cm$^3$ are simultaneously achieved, which is superior to other antiferroelectric ceramics and dielectric ceramics. By limiting the doping window, the commensurate modulation characteristics of polarization order can be maintained, which ensures the maximum polarization. A highly differentiated octahedra rotational distortion yields a multi-stage antiferrodistortion-ferrodistortion transition and a coexistence of polymorphic ferroelectric phases, significantly prolonging the polarization process. The proposed structural order differentiation shows guiding significance for the development of antiferroelectric, and the obtained energy storage performance promotes the practical applications of antiferroelectric ceramic capacitors.

Antiferroelectric (AFE) ceramics are expected to be widely employed in high currents and short pulses related pulse power fields such as hybrid electric vehicles and strong magnetic pulses, further improving their energy storage capacity is one of the current research hotspots[1–3].

Antiferroelectric ceramics are usually divided into lead-free and lead containing, the former is mainly composed of silver niobate or sodium niobate while the latter is mainly divided into lead zirconate, lead hafnate, lead lutetium niobate, etc[4,5]. From the current research

[1]Key Laboratory of Advanced Civil Engineering Materials of Ministry of Education, Functional Materials Research Laboratory, School of Materials Science and Engineering, Tongji University, Shanghai, China. [2]Max Planck Institute for Solid State Research, Stuttgart, Germany. [3]Shanghai Institute of Ceramics, Chinese Academy of Sciences, Shanghai, China. [4]School of Chemistry and Material Science, Hangzhou Institute for Advanced Study, University of Chinese Academy of Sciences, 1 Sub-lane Xiangshan, Hangzhou, China. [5]These authors contributed equally: Guanglong Ge, Jin Qian. ✉e-mail: hgwang@fkf.mpg.de; Hutengfei@mail.sic.ac.cn; apzhai@tongji.edu.cn

evaluations, lead-free and some lead containing AFEs always face high material costs, complex preparation processes and less satisfactory performance, lead zirconate based AFEs are still the primary market choice for energy storage ceramic capacitors[6]. For lead zirconate, it is firmly believed that an electric field trigged antiferroelectric-ferroelectric phase transition distinguishes a high polarity and a low polarity state, the polarization switching between them is the origin for AFE ceramics to store and release electrical energy[1-3,6]. This kind of phase transition involves two states of AFE and induced ferroelectric (FE), as well as a dynamic AFE-FE switching process. To date, one has known much about the regulation strategy of AFE states, and numerous reports have confirmed that its impact on energy storage performance is clearly effective[2,3,6-8]. However, it is still difficult to imagine how to regulate the properties of FE states and how to optimize the dynamic AFE-FE switching process, it is rather attractive and challenging to provide a modulation strategy.

Both of the induced FE state and the dynamic phase transition are derived from the AFE ground state. The room temperature lead zirconate exhibits a typical eight fold supercell structure with $D_{2h}$ spatial symmetry, the space group is $Pbam$[9,10]. Within the orthorhombic supercell, two kinds of ordered structures of polarization order and structural order coexist. The structural order represents an antiferro-distortive (AFD) mode oxygen octahedra arrangement ($a^-a^-c^0$) with antiphase rotation along the [210] and [2$\bar{1}$0] axes of one orthorhombic cell, where adjacent oxygen octahedra connects each other to form a structural skeleton[9]. The polarization order represents an antiparallel arranged lead cations in adjacent unit cells along the [100] and [$\bar{1}$00] directions, which exhibits the antiferroelectricity of lead zirconate[10]. Thus, the room temperature antiferroelectric lead zirconate always shows four fold commensurate modulated (CM) lead ions sequence, and the AFD mode octahedra superstructure stabilizes the polarization order[11]. Under an electric field, we have already known that the structural order responds to the electric field ahead of the polarization order, and the spatial position of oxygen in the octahedra shifts, which leads to a distortion mode transition from AFD to ferrodistortive (FD)[9]. The formation of FD structure will break the spatial inversion symmetry of centrally symmetric $D_{2h}$ lattice and endows the ceramic with a certain net polarization. When exceeding the AFE-FE phase critical electric field ($E_{AF}$), the antiparallelly arranged lead dipoles

configuration transforms into parallel arrangement, resulting in a sudden increase in the net polarization of the ceramic. In Fig. 1, we illustrate this process and label the AFD and FD modes as green and light blue, while labeling the FE state ($C_{2v}$) as dark blue even though its oxygen octahedra does not rotate anymore[9].

The entire process of AFE-FE phase transition is simply presented and detailed as a gradual evolution of AFE/AFD-FD-FE. This is a rather interesting process, where the stability and instability of the anti-ferroelectricity of lead zirconate are strongly related to the distortion of the oxygen octahedra, at least it is fully appropriate in CM anti-ferroelectrics. This strong correlation has led to some of our speculations. We know that within the entire orthorhombic supercell, there is no significant difference between octahedra except in spatial position although their rotation directions is antiphase. The effect of electric field on oxygen anions in octahedra is essentially comparable, and the FD phase evolves almost simultaneously in limited short time resolution when a critical electrical energy is reached. Therefore, assuming that changing the rotation angle of some of these oxygen octahedra, i.e., constructing an octahedra set with various degrees of distortion from large to small, it will be rather cool in this way to regulate the AFD-FD transition process. This idea can be well described in Fig. 1, where we define green as the standard distortion degree, such as $R_{25}$ mode of -6.6° along both the [210] and [2$\bar{1}$0] axes[12]. Meanwhile, octahedra with distortion degrees higher and lower than this value are respectively marked in red and light green. It should be noted that the four period modulated lead dipoles ordering configuration is the main contributor to high polarization after phase transition, and we are unsure whether the incommensurate modulated (ICM) structure will undergo the same AFE/AFD-FD-FE phase transition, it is necessary to maintain the lead sequence as four fold modulated here. When constructing this structure, it is easy to imagine the entire process of AFE-FE phase transition. We define this structure as a CM AFE structure with differentiated structure order, which is very special. A narrow-window differentiation of structure order does not affect the modulation of polarization order, this has been confirmed in this research. Obviously, the differentiation of structural order promotes a multi-stage AFD-FD phase transition process, the polarization growth of AFE lead zirconate is significantly prolonged (green curve in Fig. 1). At the same time, maintaining a CM structure did not reduce the maximum polarization

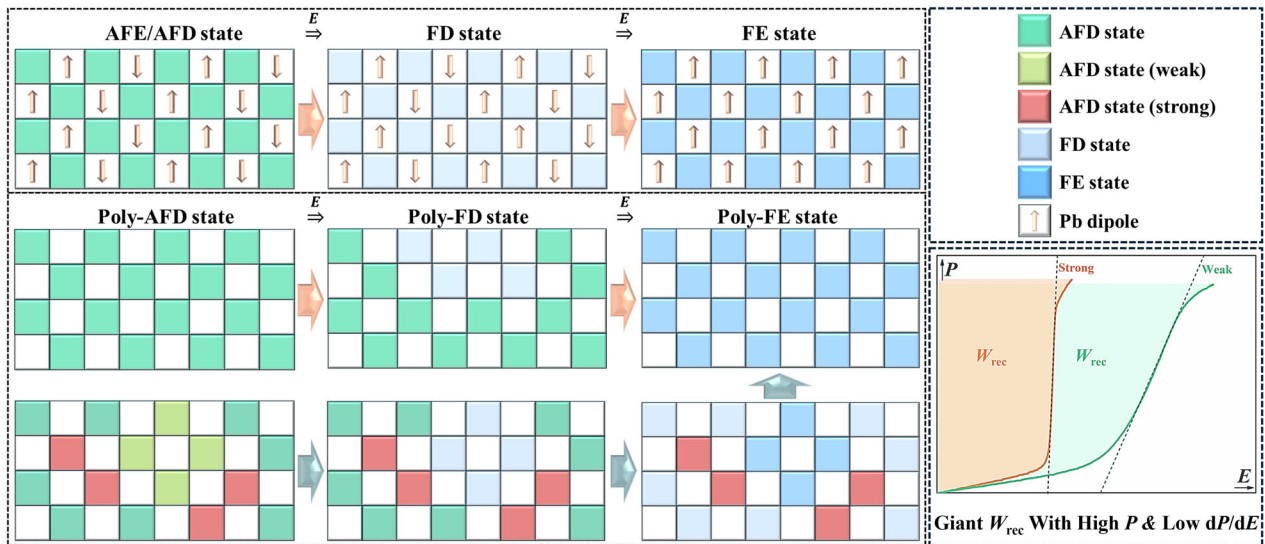

**Fig. 1 | Structural order differentiation engineering.** Schematic diagram of weak antiferroelectric-ferroelectric first-order phase transition controlled by structural order differentiation engineering, in which the AFD state represents oxygen octahedra distortion mode of antiphase rotation with a tilt angle of about 9.3° along

[100] direction[12], the weak and the strong AFD states respectively represent distortion modes with smaller and higher tilt angles, and the darkening of FD color blocks indicates a reduced octahedra distortion.

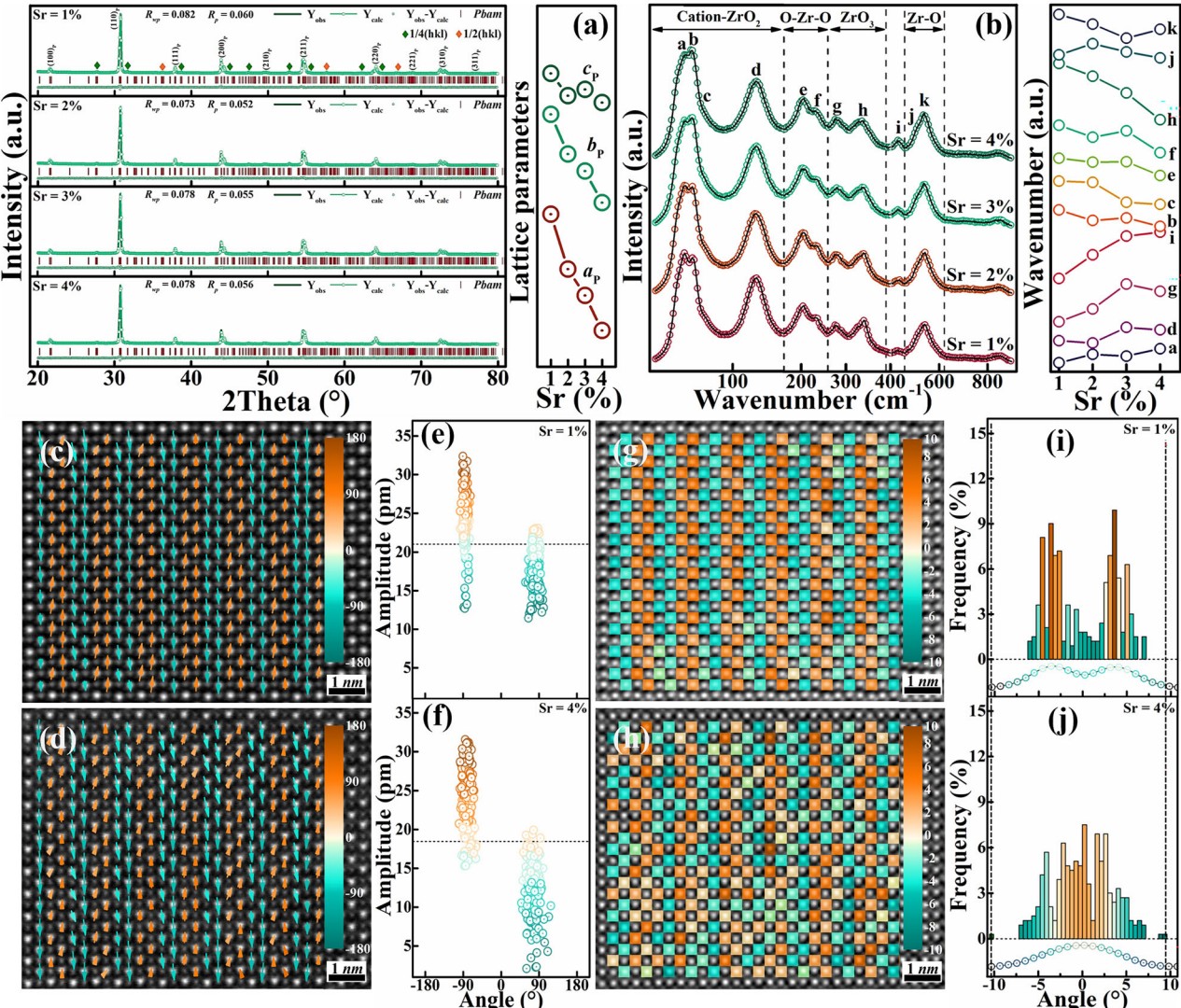

**Fig. 2 | Visualization of structural order differentiation. a** The refinement of room temperature phase structure of antiferroelectric S1 ~ S4. **b** The room temperature Raman spectra and the changes in Ramanshift of several wavebands. **c, d** The [001]$_P$ projected high-resolution lattice images and the lead displacement vector maps overlaid on the HAADF-STEM images of antiferroelectric S1 and S4, in which the horizontal and vertical directions respectively point towards the [$\bar{1}$10]$_P$ and the [110]$_P$ directions of a pseudo-cubic lattice. **e, f** The magnitude and angle distribution of the corresponding lead ion displacement vector. **g, h** The [001]$_P$ projected high-resolution lattice images and the oxygen octahedra rotation maps of antiferroelectric S1 and S4. **i, j** The corresponding rotation angle distribution of oxygen octahedra.

of the ceramic, and the energy storage density of the ceramic was significantly improved. That is, we have proposed an novel strategy that can both regulate dynamic AFE-FE phase transition and directly affect the polarization of the induced FE states. This study provides reference for further understanding the essence of lead zirconate, and promotes the development and application of next-generation high-performance lead containing antiferroelectric ceramic capacitors.

## Results and discussion

To construct structural order differentiated commensurate modulated antiferroelectric, we adopt a basic strategy of compressing the volume of antiferroelectric cells. We know that the cell volume effect in lead zirconate is always related to the critical phase transition electric field[13,14]. However, our perspective about lattice tuning effect here is different, we introduced a high valence cations with a much smaller ionic radius into the lead occupied position, which will induce significant distortion by enhancing the lattice anisotropy in its vicinity. Then, a slightly smaller divalent cation was applied to amplify the

anisotropic effect, the ion occupancy in a larger lattice range in three-dimensional directions will be modulated. Specifically, we chose the A-site Y$^{3+}$ and Sr$^{2+}$ co-doping method to differentiate the structural order. The ionic radius of Y$^{3+}$ (1.19 Å, CN = 12) is much smaller than that of Pb$^{2+}$ (1.49 Å, CN = 12)[2,15], which can easily cause high distortion at the lattice points and is the initiator of inducing structural distortion. The ionic radius of Sr$^{2+}$ (1.44 Å, CN = 12) is slightly smaller than that of Pb$^{2+}$[8,13], and it can be used as an enhancer to slowly amplify the effect of distortion to differentiate the structural order. It should be noted that we need to maintain the basic CM AFE structure, which requires a limited doping window. The doping content we chose is relatively small, with the initiator content controlled at 1 mol% and the enhancer content controlled at 0 ~ 4 mol% (See the Methods). As a result, in terms of the average structure, the refined XRD profiles reveal a typical perovskite structure as shown in Fig. 2(a)[2,3]. Most of the main Bragg diffractions are splitting, indicating a low symmetry. Outside of the main Bragg diffractions, there are a large number of superstructure diffractions representing the ordered structure of AFE lead

zirconate[12,16]. Among them, the green diamond represents $\Sigma$-type reflections, such as $(112)_O//(\frac{3}{4}, \frac{-1}{4}, 1)_P$ diffraction near 28° and $(310)_O//(\frac{7}{4}, \frac{-5}{4}, 0)_P$ diffraction near 47° (O and P refer to orthorhombic and pseudocubic cells respectively), which is related to the polarization order of AFE lead zirconate[12,16]. The yellow diamond represents R-type reflections and belongs to the AFD ordered structure, such as $(141)_O//(\frac{3}{2}, \frac{1}{2}, \frac{1}{2})_P$ diffraction near 36° and $(261)_O//(\frac{5}{2}, \frac{1}{2}, \frac{1}{2})_P$ diffraction near 57° [3,12]. For a convenient description, we will only use pseudocubic unit cells for diffraction indices in the following, and the superstructure diffractions are respectively written as $1/4(hkl)$ and $1/2(hkl)$. Overall, the Sr/Y co-doped system (PSY) exhibits diffraction characteristics and spatial symmetry consistent with pure lead zirconate (Pbam)[2,3,12,16]. At the same time, with the increase of Sr doping content, the unit cell gradually shrinks having a decreased lattice parameter set of $a_P$, $b_P$, and $c_P$ from 4.1190 Å to 4.0896 Å, 4.1179 Å to 4.0892 Å, and 4.1174 Å to 4.0893 Å. In addition to XRD profiles, we also considered the vibration modes of the local structure of the PSY system modulated by the initiator and enhancer co-substitution. As Fig. 2(b), the local structure of the PSY system exhibits similar characteristic vibrations, indicating that the selected doping window is appropriate. Throughout the entire spectral range, we marked over ten featured vibration bands labeled as a ~ k and divided them into four segments[17]. These bands are generally consistent with those observed in PbZrO₃[3,8,14,17,18]. In the lowest frequency region (<150 cm⁻¹), attributed to the cations-octahedra lattice modes, bands a to d represent two trends in Ramanshift[19]. It can be postulated that the blueshift of bands a (45 cm⁻¹) and d (127 cm⁻¹) indicates an increased cation-anion interaction, while the redshift of bands b (55 cm⁻¹) and c (69 cm⁻¹) should be related to the restricted movement of cations. The redshift of bands e (205 cm⁻¹) and f (235 cm⁻¹) indicates a weakened bending vibration of O-Zr-O. The Zr-O bond stretching vibration modes of bands j and k undergo redshift with Sr doping, indicating potentially an expand BO₆ octahedra. For the BO₆ torsional modes, bands g and h always show opposite behavior[3,8,14,20], we may consider it as an anisotropic response of octahedra distortion in a Cartesian coordinate. Thus, band g perhaps represents a degenerate mode in a two-dimensional plane such as the $(00l)$ planes, while band h should be a component in a vertical direction. The overall structure exhibits spatial inversion symmetry of Pbam, and the co-doping of initiator and enhancer leads to an increased antiferroelectric stability with a complex octahedra tilt in certain directions.

More insight of the structural order can be well drawn from the atomic-resolution HAADF-STEM images. From the [001] zone axis, the atomic-scale thin lead dipole patterns with clear angle contrast are well depicted in Fig. 2c, d. We have known that the lead dipole sequences of commensurate modulated antiferroelectric lead zirconate always show a featured four-fold head-to-tail configurations, where the displacement directions of adjacent lead cations are along the [110] and [1̄10] directions of its pseudocubic unit cell[9,10]. Here it is clearly observed that PSY system maintains a up-up-down-down type commensurate modulated polarization ordered structure wthin this doping window, which confirms to our expectations. The average amplitude of lead ion displacement in antiferroelectric S1 is about 21.0 pm, while that of antiferroelectric S4 is slightly decreased to around 18.5 pm. The magnitude distribution range of lead dipole sequences in antiferroelectric S1 is wider but it is mainly concentrated at angle of -90°, it is parallel to the [110] or [1̄10] direction. However, by combining the lead dipole sequence distributed at around 90°, an almost fully compensated modulation structure can be achieved. As for antiferroelectric S4, its lead dipole configuration distribution is almost symmetrical in terms of angle and magnitude range, it is also easy to imagine an almost fully compensated modulation structure. As shown in Fig. 2g, the octahedral sequence of antiferroelectric S1 also exhibits a four-fold lattice period commensurate modulated mode, characterized by a set of green-green-yellow-yellow type octahedral

network with different rotation directions. However, the distribution pattern of oxygen octahedra in antiferroelectric S4 is characterized by short period or localized four-fold corner connections, while the octahedral distribution in other regions is rather chaotic, as Fig. 2h. By analyzing the distribution of octahedral rotation angles of antiferroelectrics S1 and S4, we found that the latter possesses a much wider range of octahedral rotation angles than the former (-10.3 ~ 9.4° VS. -6.0 ~ 6.9°). It should be noted that the rotation angle of pure lead zirconate is larger than that of antiferroelectric S1[12], the oxygen octahedral rotation angle of Sn⁴⁺ doped lead zirconate itself should be limited. In antiferroelectric S1, the distribution frequency map of octahedra rotation angles is basically symmetrical relative to zero degree rotation, showing a bimodal distribution pattern located in the positive and negative rotation angle regions (±3.6°), as displayed in Fig. 2i. Of course, there is also a certain amount of distribution probability within a small angle range close to zero degree rotation. However, for antiferroelectric S4, the situation is completely different. We can easily observe that the rotation angle of its oxygen octahedra follows a Gaussian like distribution, resulting in a mixture of high and low rotation angle octahedra configuration. This means that for antiferroelectric S1, its oxygen octahedra pattern is generally consistent with a pure lead zirconate with only a portion of the regions that has undergone structural order differentiation showing a higher or smaller rotation angle. However, the structural order differentiation in antiferroelectric S4 is rather severe, of which not only weakly distorted octahedra (angle amplitude approaches 0°) grow in its lattice but also many strongly distorted octahedral (angle amplitude is higher than 4°) rotation modes exist. As a result, the structural order differentiation engineering envisioned in Fig. 1 has been achieved, and the structure still maintains the polarization order characteristics of commensurate modulation having the average spatial symmetry of Pbam, the corresponding lattice can be drawn by stacking it in three-dimensional space in the same way as Fig. 1. After constructing the structure we need, the impact of this structural order differentiation engineering on the polarization-depolarization response of commensurate modulated antiferroelectric ceramics needs to be revealed.

In our assumption, further differentiation of structural order will promote multi-stage AFD-FD transition, and the process of AFE-FE phase transition will be significantly prolonged. A prolonged AFE-FE phase transition means that the polarization-electric field (P-E) response relationship will change from a strong first-order to a weak first-order phase transition, this will appear as the slanted like P-E loops. As Figure S1, the square like double "S" type P-E loops of $Pb_{1-1.5x}Y_xZr_{0.6}Sn_{0.4}O_3$ system begin to exhibit a slightly slanted with Y³⁺ doping (reduced $dP/dE$). The $Pb_{1-1.5x}Y_xZr_{0.6}Sn_{0.4}O_3$ system doped with 1 mol% Y³⁺ shows low hysteresis and high polarization at the same time, it is selected as the optimal matrix, and Y³⁺ is defined as initiator. As depicted in Fig. 3a and Figure S2, the Sr doping window is selected to be 0 ~ 4 mol%, a higher contents resulting in a commensurate AFE-polymorphic modulated AFE phase transition. Sr²⁺ is defined as enhancer as it further enhances the initiator prolonged polarization process. With the enhancer Sr²⁺, the P-E loops further transform from square to slanted shape, still exhibiting large hysteresis (50 ~ 70 kV/cm) and high polarization (>50 μC/cm²). In the PSY system, the cell volume effect also enhances the antiferroelectric stability, as Fig. 3e and S2. However, this research mainly focuses on its tuning effect on structural order and the modulation of polarization-depolarization response caused by it. As shown in Fig. 3f, via structural order differentiation engineering, the polarization growth rate with the electric field significantly decreased from 1.59 to 0.41 and then to 0.24. Comparing antiferroelectric S1 and S4, we know from the above analysis that their structural difference lies in the latter having a more significantly differentiated structural order, so what is the mechanism by which this differentiated structural order prolongs the polarization process? To this, we studied the thermodynamic evolution of the PSY system

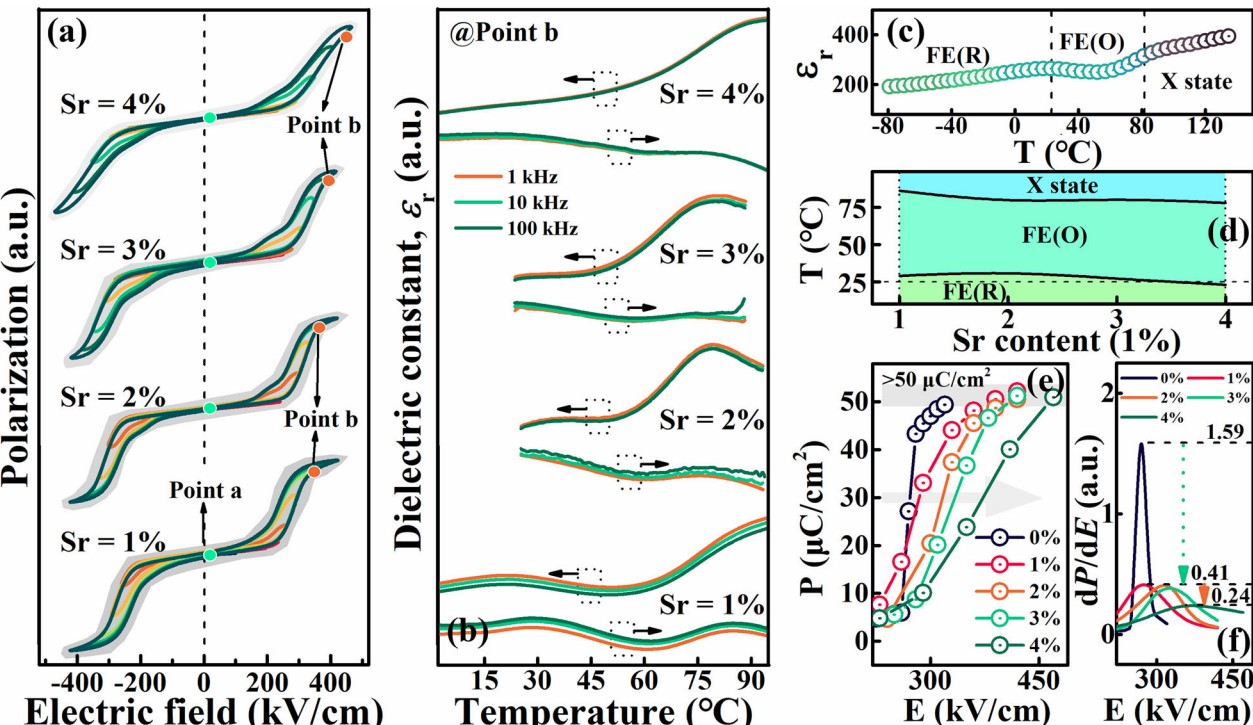

**Fig. 3 | Thermodynamic evolution of antiferroelectric and ferroelectric states.** **a** The room-temperature *P-E* loops of antiferroelectric S1 - S4, in which point a and b represent antiferroelectric and ferroelectric state respectively. **b** The dielectric-temperature response of the induced ferroelectric state of antiferroelectric S1 - S4. **c** Schematic diagram of phase region division based on the dielectric constant-temperature response relationship. **d** Structure-composition phase diagram of the induced ferroelectric states versus temperature. **e, f** The polarization-electric field relationship and the $dP/dE − E$ curves of antiferroelectric S0 - S4.

before and after the AFE-FE phase transition. The method we applied was to induce the AFE-FE phase transition by applying a bias voltage, and then observe the temperature depedent response relationship of different states under different electric fields, the results are shown in Fig. 3b and Figure S3. When the applied bias electric field is about 0 kV/cm (point a in Fig. 3a), i.e. in the AFE ground state, the PSY system exhibits a similar dielectric temperature response relationship ($\varepsilon − T$ curves) with no obvious frequency dependent dielectric dispersion, the dielectric imaginary parts (dielectric loss) at room temperature of antiferroelectric S1 - S4 are all around 0.5%. By analyzing the in-situ temperature depedent XRD patterns or Raman spectra, it is easy to conclude that PSY system generally exhibits a temperature induced phase transition process of CM AFE $\xrightarrow{T_0}$ ICM AFE $\xrightarrow{T_1}$ multicell cubic $\xrightarrow{T_2}$ paraelectric. We have provided a detailed analysis of the corresponding details in our previous research[3,8,14,20], and we only present the relevant experimental evidence and conclusions here (Figure S4 and S5).

When a bias electric field greater than zero is applied, the curve of the PSY system undergoes significant changes. See Figure S3, here we employ antiferroelectric S3 as an example to illustrate this change, its critical $E_{AF}$ is about 365 kV/cm. We divide the *P-E* loops into AFE stage and FE stage. We know that applying an electric field will increase the voulume of AFE cell, thereby the AFE phase stability gradually decreases and eventually transforms into a FE crystal cell when exceeding $E_{AF}$. Therefore, in the AFE stage, as the bias electric field increases, the stability of the AFE phase gradually decreases, and it can be observed that the critical temperature of the CM AFE-ICM AFE phase transition significantly decreases from 146 to 45 °C. When the electric field is further increased to about 410 kV/cm, entering the FE stage of the *P-E* loops, the $\varepsilon − T$ curve of the induced FE state is completely different from its AFE state. The significant increase in the imaginary part of the dielectric curves at room temperature from 0.5%

to 5% is one of the key features in determining the formation of FE states. Within the temperature range of 23–90 °C, two dielectric anomaly peaks can be determined based on the real or imaginary part of the dielectric curves. Moreover, no significant frequency dispersion was observed throughout the entire temperature range, indicating that the induced FE state of antiferroelectric S3 is a conventional FE phase. In addition, we also know that electric fields enhance the interaction between ferroelectric dipoles, the stability of FE states will increase with the bias electric fields. Taking antiferroelectric S1 and S4 as examples, it is evident that as the electric field increases, the critical temperature of their dielectric anomaly peaks shift towards higher temperatures. Since the critical phase transition temperatures of both AFE and FE states change with increasing electric field, how can we compare the differences in induced FE states of the PSY system with different enhancer doping content? We propose to determine the electric field required for all compositions based on a near saturated standard polarization here. The standard polarization we have chosen is around 50 μC/cm², which has basically reached the saturation polarization of PSY system.

According to this, the electric fields required for antiferroelectric S1 - S4 ceramics are about 370, 390, 405, and 470 kV/cm respectively. It should be noted that the bias electric field used for some compositions may deviate slightly from their standard electric field. This does not affect the subsequent analysis, and the trend changes caused by this deviation are easily imaginable. Beyond this, we also define the two dielectric anomaly peaks appearing in the bias depedent $\varepsilon − T$ curves as $FE_R \xrightarrow{T_{R-O}} FE_O$ and $FE_O \xrightarrow{T_{O-X}} X$ state respectively as shown in Fig. 3c, where R and O respectively indicate rhombohedral and orthorhombic symmetry, and X state represents possible paraelectric or other ferroelectric phases. It also should be noted that if there are obvious dielectric loss peaks, we prioritize using loss peaks for locating temperature. The determination of symmetry will be provided in the

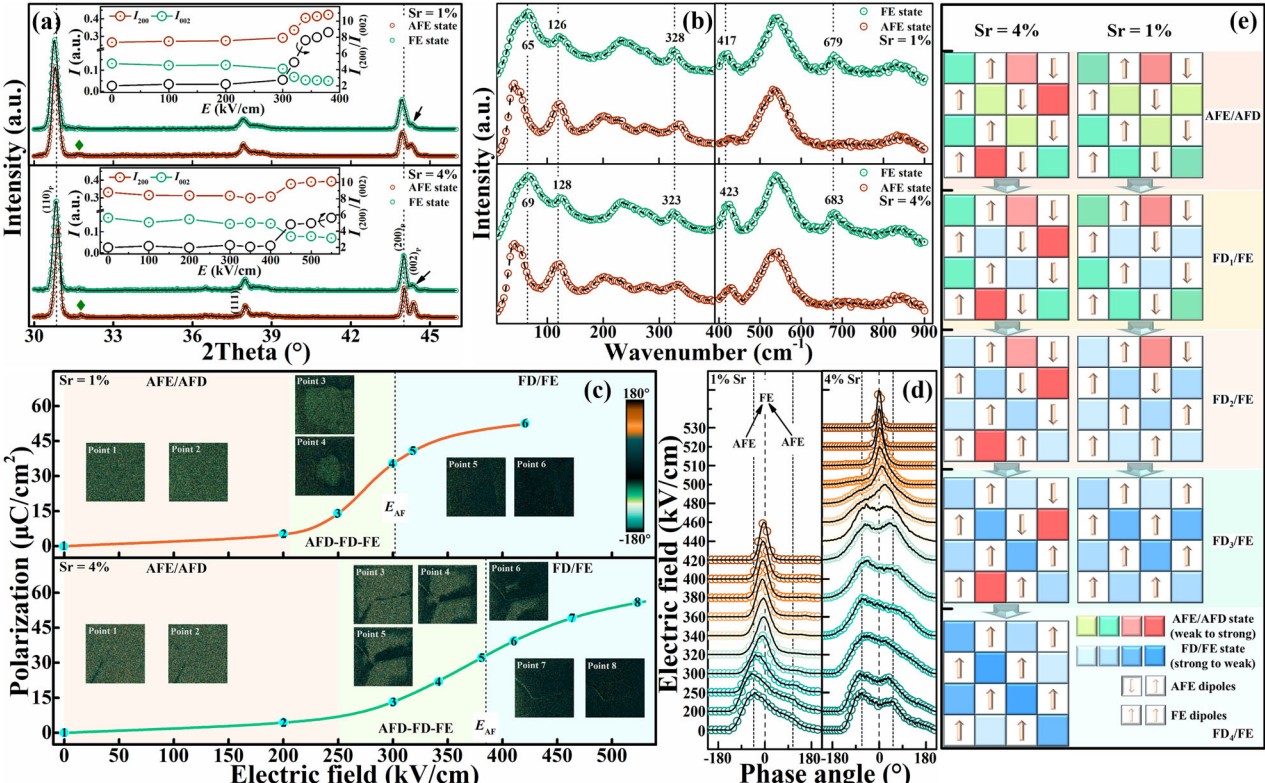

**Fig. 4 | Dynamic process of phase and domain structure. a** The in-situ XRD patterns versus electric field of S1 and S4 respectively, in which the illustrations indicate the trend of the peak intensity of the main Bragg diffraction versus the electric field. **b** The electric field-dependent Raman spectra of antiferroelectric S1 and S4. **c, d** The PFM images and the corresponding phase angle distribution under different bias electric field. **e** Schematic diagram of the details of antiferroelectric-ferroelectric phase transition.

following. Based on this definition, we have plotted the composition-temperature phase diagram of the induced FE state of the AFE PSY system, as shown in Fig. 3d. In fact, it is not difficult to find that for antiferroelectric S1 ~ S4, their critical temperatures $T_{R-O}$ are all around room temperature. Considering the influence of electric field devia-tion, the critical temperature $T_{R-O}$ of antiferroelectric S3 will be slightly lower than 27 °C, while that of antiferroelectric S4 will be slightly higher than 23 °C. We can assume that their actual critical temperatures are both very close to 25 °C, which is a suitable but not important value and does not affect our evaluation of antiferroelectric S1 and S4. According to Fig. 3b, it is clear that the dielectric anomaly peak $FE_R \xrightarrow{T_{R-O}} FE_O$ of antiferroelectric S1 ~ S4 all exhibits diffuse phase transition characteristics. For a diffuse phase transition, the phases on both sides of the dielectric anomaly peak will disperse into each other, generally causing a multiphase coexistence state. Moreover, with the Sr doping content, the dielectric anomaly peak becomes more and more flat, which is considered as a gradually improved degree of dif-fusion. Therefore, both considering the critical temperature and the diffusion degree, we can assume that the $FE_R$ phase dominates in antiferroelectric S1, the $FE_O$ phase dominates in antiferroelectric S2 and S3, while the $FE_R$ and the $FE_O$ phases in antiferroelectric S4 are both dominant. Multiphase coexistence often leads to a complex polarization direction combination, which will facilitate the polariza-tion, i.e. a high maximum polarization ($P_{max}$) than that of PbZr$_{0.6}$Sn$_{0.4}$O$_3$ matrix. The coexistence of multiple phases will pro-mote polarization, but their competition will suppress polarization response and prolong the polarization growth process. Therefore, it is not difficult to understand that there are dominant phases in anti-ferroelectric S1 ~ S3, the polarization process needs to prioritize the dominant phase and the polarization delay effect is slightly

insufficient. In antiferroelectric S4, $FE_R$ and $FE_O$ phases competing with each other will cause a sufficient delay in polarization response, resulting in a significantly reduced $dP/dE$. Especially considering the structural order differentiation engineering, highly differentiated antiferroelectric S4 naturally exhibits a special ferroelectric coex-istence phase state, which not only endows it with high $P_{max}$ but also delays its polarization-electric field response. Moreover, considering the difference in rotation angles between octahedra, the coexistence of $R/O$ symmetric FE states should be attributed to a AFE unit cell set with stronger or weaker rotation distortion octahedra. Therefore, several FE states can be observed in lead zirconate[9,21,22], and the attri-bution of its FE state should be strongly related to the applied electric field because its FE state should have a similar thermodynamic response relationship as the PSY system.

A diffuse phase transition contributes to a possibly coexisting ferroelectric states. To this, we applied an electric field depedent XRD to observe this phenomenon, the results are well depicted in Fig. 4(a) and Figure S6. Overall, as the electric field increases, all diffraction peaks in the XRD profiles of antiferroelectric S1 and S4 shift towards lower angles, indicating an electric field induced cell expansion. For a convenient description, we have calculated the changes in intensity of the (200)/(002) indexed diffractions of antiferroelectric S1 and S4 with the electric field, as shown in the illustration in Fig. 4(a). We know that determining the AFE-FE phase transition requires focusing on the peak shape of main Bragg diffractions and the intensity of featured super-structure diffractions (green diamond). It is obvious that the diffrac-tion intensity of the (200)/(002) indexed crystal planes change significantly near the critical electric field $E_{AF}$. During this process, the $(\frac{5}{4}, \frac{-3}{4}, 0)$ superstructure diffraction also gradually disappears. The dipoles with up-up-down-down configuration related to lead

displacement form a special periodicity in structure, i.e. super-structure. This superstructure changes with the electric field and transforms into a up-up-up-up configuration ferroelectric dipole sequence near the critical electric field. At this point, the diffraction of the superstructure disappears, indicating the breaking of the AFE polarization order. In addition, the important signal indicating the occurrence of phase transition is the change in the main Bragg diffraction. What we are familiar with is that the $(h00)_P$ indexed main Bragg diffraction can serve as a significant difference between orthorhombic and rhombohedral symmetry pervoskites. Related to the interplanar spacing, the Bragg diffraction has different peak shapes in orthorhombic and rhombohedral occasions, the former is bimodal or trimodal while the latter is unimodal. Especially in the case of perovskite lead zirconate, the $(h00)$ indexed main Bragg diffraction of an orthorhombic crystal splits into two peaks, and the intensity of the low angle peak is about twice that of the high angle peak. Thus, for antiferroelectric S1, the intensity of $(200)$ diffraction suddenly increases by about 40%, while that of $(002)$ diffraction suddenly decreases by about 65%. For antiferroelectric S4, its $(200)/(002)$ diffraction intensity have changed by about 20% and −50% respectively. Therefore, the diffraction intensity ratio of $I_{(200)}/I_{(002)}$ of antiferroelectric S1 significantly increases from 2 to 8 before and after the AFE-FE phase transition, while that of antiferroelectric S4 increases from 2 to 6. Considering the differences in diffraction between $O$ and $R$ symmetric crystal cells, it can be concluded that the FE state of antiferroelectric S1 exhibits more pronounced $FE_R$ characteristics. Meanwhile, due to the presence of $(002)$ diffraction, the FE states of both antiferroelectric S1 and S4 are $FE_O$ and $FE_R$ coexisting polymorphic FE states, which is consistent with the essence of a diffuse phase transition. The variation of the vibration characteristics of local structures during the AFE-FE phase transition can also confirm the above viewpoint, as Fig. 4(b) and Figure S7. We know that the typical sign of the appearance of $FE_O$ phase ($Cm2m$) is the sudden emergence of new vibration modes near 412 and 673 cm$^{-1}$ [14,20]. Therefore, both antiferroelectric S1 and S4 exhibit this characteristics, indicating that their FE states are composed of $FE_O$ phase. Meanwhile, we also know that the enhanced spectral intensity in the low frequency region (<30 cm$^{-1}$) is one of the typical sign of $FE_R$ phase [23,24]. And it is evident that the spectral intensity of antiferroelectric S1 and S4 suddenly increases in this frequency range after entering their FE state [14]. In addition, for the $Pbam - Cm2m$ phase transition, multiple vibration modes including bands c - g and k will not show apparent changes in Ramanshift, while band h shows a significant redshift and the intensity of band e is significantly decreased. The AFE-FE phase transition of antiferroelectric S1 and S4 basically displays the above characteristics except for the band e. For a $FE_R$ phase ($R3c$), this band does not exist anymore. Therefore, the FE states of antiferroelectric S1 and S4 should be at least composed of $FE_O$ and $FE_R$ phases. Moreover, the $FE_O$ phase in the antiferroelectric S4 exhibits a higher vibration frequency in terms of characteristic vibration modes, which may mean that stronger $FE_O$ phases are derived from the highly distorted portion of the antiferroelectric unit cell set with significantly differentiated octahedral rotation sets, which is what antiferroelectric S1 lacks. Therefore, related to the critical temperature $T_{R-O}$ and the diffusion degree, there is a dominant $FE_R$ phase in antiferroelectric S1, which makes it difficult for the $FE_O$ phase to compete with it and causes a balance like polymorphic FE coexistence. This kind of balanced manner limits the promoting effect of multiple polarization directions on polarization growth but disrupts their delaying effect on polarization evolution. Instead in antiferroelectric S4, both $FE_O$ and $FE_R$ component are dominant and can compete with each other to sufficiently prolong the polarization process.

The delay effect of structural order differentiation engineering on AFE-FE phase transition can be visually demonstrated through bias piezo-response force microscope (PFM). As we all know, PFM is a classic technique for detecting the domain structure of piezoelectric/ferroelectric ceramics, by means of using a AC signal to detect the phase angle of the domain structure after applying DC voltage to polarize the sample [25,26]. It should be noted that currently we cannot use both AC and DC to polarize the sample and detect the domain structure at the same time. We know that the FE states derived from AFE ceramics cannot exist stably and will rapidly depolarize to their original non-polar state if the electric field is removed. Therefore, PFM seems to be not applicable for detecting AFE-FE phase transition. However, we propose that the AFE-FE phase transition can be driven by applying a horizontal bias electric field and the induced FE state can be stabilized by maintaining a bias. Thus, we sputter interdigital electrodes on the well-polished surface of the ceramics and apply a constant horizontal DC bias electric field. The finger gap should be controlled within 3 - 5 μm, and the ceramic thickness needs to be much greater than this value to avoid leakage current in the thickness direction of the ceramics. In addition, considering that the AFE-FE phase transition may be completed within an narrow bias window, it is necessary to collect signals at the intersection of multiple grains including grain boundaries, the grain boundaries can serve as landmarks for distinguishing phase transition areas. As Fig. 4c and Figure S8-S10, we selected a 2 μm*2 μm area and detected the surface distribution of phase angles of antiferroelectric S1 and S4 under different electric fields. We defined FE domains here as contiguous areas with similar phase angles. We divide the AFE-FE phase transition into the AFE/AFD stage, the AFD-FD/FE stage, and the final FD/FE stage, which fully demonstrates the details of this phase transition. For both antiferroelectric S1 and S4, it is obvious that we can only probe negligible domain morphology when in their AFE/AFD stages. In terms of phase angle distribution, both antiferroelectric S1 and S4 exhibit a bimodal type phase angle distribution as Fig. 4d, which indicates the AFE nature of the up-up-down-down type polarization configurations. And it should be noted that due to spatial resolution, the PFM phase distribution of the AFE/AFD stage may differ from the results of HAADF-STEM. When entering the AFD-FD/FE stage, clear phase angle contrast begins to appear in both antiferroelectric S1 and S4, and two different regions colored with dark green and light green appear in the PFM images. According to our definition, the light green regions still maintain AFE characteristics, while the dark green regions should be considered to display textured FE domains because their phase angles are similar. Especially, these FE domains seem to be more prone to nucleation at grain boundaries. At this stage, FE domains grow with the increase of electric field, accompanied by a certain enhancement of polarization. The phase angle distribution still shows the same bimodal characteristics as the AFE/AFD stage, while the peaks in the positive phase angle quadrant gradually increase, indicating the coexistence of FE domains and AFE areas. When the applied bias electric field exceeds $E_{AF}$, the growth of FE domains is basically completed and occupies the entire area. At this stage, the bimodal phase angle distribution transforms into a single peak, indicating that the phase angle within the entire area has begun to orient. Finally, further increases the electric field, this single peak gradually narrows, indicating that the phase angle in the entire area has been highly oriented, and the FE domains gradually become oriented with the electric field after the growth is completed. Comparing the domain evolution patterns of antiferroelectric S1 and S4, it is not difficult to find that the AFD-FD/FE stage of antiferroelectric S1 is much shorter. Moreover, after entering the FD/FE region, a portion of areas of antiferroelectric S4 still remain AFE characteristics. This indicates that the AFE-FE phase transition and the polarization evolution process of antiferroelectric S4 are significantly diffuse. At a sufficiently high electric field, the unimodal type phase angle distribution of both antiferroelectric S1 and S4 is narrow, indicating a highly oriented FE domain, which is the reason for a high $P_{max}$. As shown in Fig. 4e, a highly differentiated structural order disrupts the AFE/AFD-FD/FE phase transition process. Especially, by inducing octahedra sets with different rotational distortions, the AFD-FD

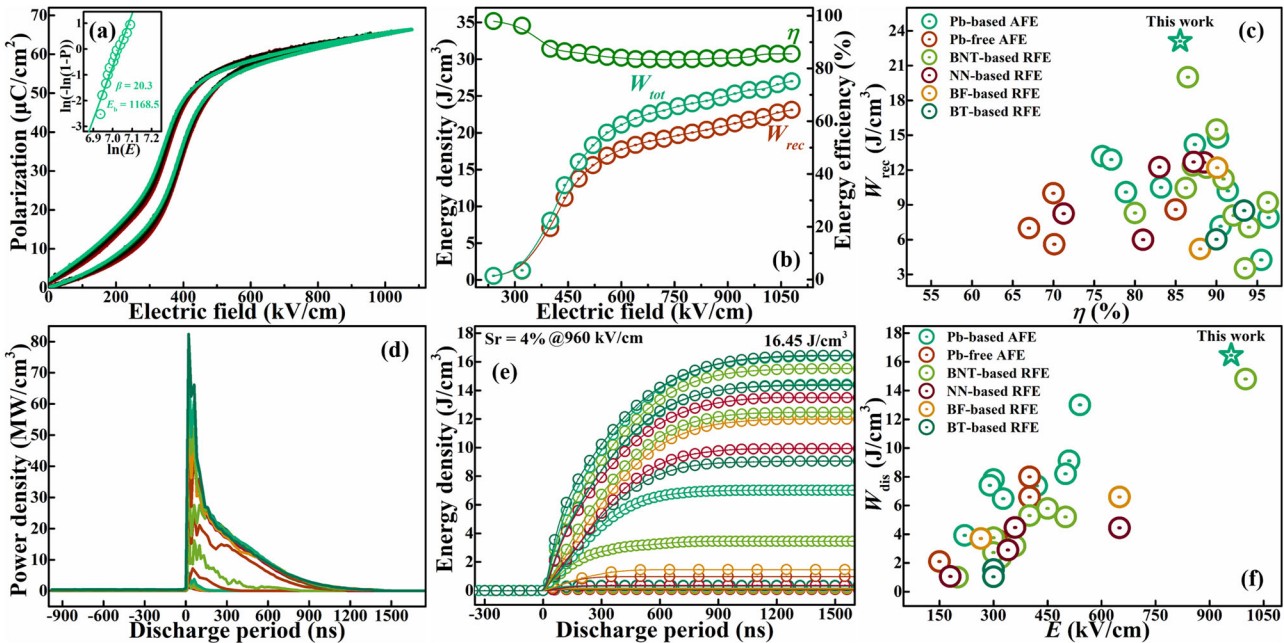

**Fig. 5 | Energy storage performance. a, b** The room temperature *P-E* loops and the energy storage properties of multilayer ceramic capacitors. **c** The comparison of energy storage performance[8,19,27–58]. **d, e** The charge-discharge curves and the discharge properties of MLCC. **f** The comparison of charge-discharge performance[19,27–42,44,45,47–58].

transition will be differentiated into multi-stage forms, significantly prolonging the polarization process. At the same time, because the modulation characteristics of the antiferroelectric polarization order were not fundamentally broken, the ceramic will still maintain a high polarization after sufficient AFE-FE phase transition.

The energy storage performance of AFE ceramics will be significantly improved by simultaneously delaying polarization process and maintaining maximum polarization, as Fig. 1. Here we calculated the energy storage performance of PSY system under the a comparable maximum polarization, as shown in Figure S2. Under the maximum polarization (about 50 μC/cm²), the required electric field increases from 320 to 470 kV/cm, which is due to the improvement of phase stability and also the polarization delay. The energy storage density ($W_{rec}$) of PSY system increased of about one-third from 10.97 to 14.63 J/cm³ with the enhancer Sr. At the same time, the energy storage efficiency ($\eta$) first decreases and then increases, reaching its maximum at about 86.00% when the enhancer content is 0 mol% and 4 mol%. According to the complex impedance spectroscopy, the phase region end composition Sr4 possesses both lowest grain/grain boundary conductivity as well as highest grain/grain boundary activation energy (Figure S11), it should perform a better breakdown performance than S0-S3 and is selected as the optimal composition when jointly considering its highly differentiated structural order and the most superior energy storage performance. To fully demonstrate its energy storage potential, the multi-layer ceramic capacitors (MLCC) were prepared, as shown in Fig. 5 and Figure S12. The effective dielectric area of the MLCC is 2 mm * 2 mm * 5 layers, and the thickness of a single layer is about 13.0 μm. By appling DC Weibull distribution, the breakdown strength of antiferroelectric MLCCs is approximately 1170 kV/cm with a modulus $\beta$ of more than 20. Under an AC electric field (10 Hz), we obtained the saturated *P-E* loops with slightly lower breakdown strength of 1080 kV/cm, as Fig. 5a. The maximum polarization at this electric field is about 66 μC/cm², and the hysteresis width keeps stable at 48 kV/cm. The dependence curves of the energy storage performance on the electric field are shown in Fig. 5b, the energy storage density and energy storage efficiency of antiferroelectric

MLCCs are 23.11 J/cm³ and 85.55%. Compared with antiferroelectric S0, the energy storage density of antiferroelectric S4 has significantly increased by about 110%, while its energy storage efficiency has decreased by 0.46%. As Fig. 5(c), the energy storage properties achieved in this work are superior to other dielectric ceramics or MLCCs[8,19,27–58]. Also, considering the pratical applications, we provide the over damped charge-discharge curves of antiferroelectric S4, as Fig. 5d. At a resistance of 330 Ω, the maximum electric field of the over damped charge-discharge tests of antiferroelectric S4 is 960 kV/cm, and the maximum power density exceeds 80 MW/cm³. Perhaps related to the prolonged polarization process, the depolarization process is also prolonged, and the discharge period of antiferroelectric S4 reaches several hundred nanoseconds, its discharge energy density can reach up to 16.45 J/cm³, as Fig. 5e. The achieved discharge energy density is also excellent as compared to other dielectric ceramics, as Fig. 5f[19,27–42,44,45,47–58]. The structural order differentiation engineering has created an antiferroelectric phase with different rotation distortion octahedra configurations and CM polarization order, which brings a multi-stage AFD-FD evolution and breaks the quasi transient AFE-FE phase transition, and ultimately endows the antiferroelectric ceramics with excellent energy storage and discharge performance.

In summary, we have proposed a strategy of structural order differentiation engineering and verified its effect in improving the energy storage performance of commensurate modulated antiferroelectric ceramics. By introducing the initiator Y, the growth rate of polarization to electric field starts decreasing. The introduction of enhancer Sr significantly broadened the distribution range of rotation angles, resulting in a mixture of high, medium and low rotation angle antiferro-distortion. The highly differentiated octahedra set further subdivides the antiferroelectric-ferroelectric phase transition by inducing multi-stage antiferrodistortion-ferrodistortion transitions. Also, it endows the commensurate modulated antiferroelectric phase with a coexistence of orthorhombic and rhombohedral symmetric ferroelectric phases, which generate more and complex polarization directions and are beneficial for the polarization growth of antiferroelectric ceramics. In addition, the stable polarization order with

commensurate modulation provides the main polarization after the phase transition, and the maximum polarization of the antiferro-electric ceramic does not decay with doping. Therefore, the energy storage and discharge performance of antiferroelectric ceramics have been significantly improved, the energy storage density has doubled compared to the undifferentiated composition, while the energy storage efficiency remains basically unchanged. The structural order differentiation engineering shows a certain guiding significance for further developing the next generation high-performance antiferro-electric materials, and the achieved ultra-high energy storage perfor-mance enhances the application potential of antiferroelectric ceramic capacitors.

## Methods
### Sample fabrication
The $(Pb_{0.985-x}Sr_xY_{0.01})(Zr_{0.6}Sn_{0.4})O_3$ (x = 0 ~ 0.04, denoting as S0 ~ S4) antiferroelectric ceramics (PSY system) were prepared via the tradi-tional solid-state reaction. Analytically pure $Y_2O_3$, $Pb_3O_4$, $ZrO_2$, $SnO_2$ and $SrCO_3$ powders were weighed according to stoichiometry with a 2% molar ratio excess of $Pb_3O_4$, and thent the mixture was ball milled with ethanol and calcined at 800 ~ 850 °C. A second ball milling was adopted to refine the powder size. The ceramic green pellets were prepared via tape-casting method[6]. The green pellets were kept in closed alumina crucible at 1250 ~ 1350 °C for 2 ~ 4 hours to obtain ceramics, a lead atmosphere compensated with sheets consisting of equal molar of PbO and $ZrO_2$ is necessary. The multi-layer ceramic capacitors (MLCCs) were prepared via tape-casting process, a high-temperature platinum slurry (Pt-75 slurry, composed of pure Pt parti-cle and $ZrO_2$ containing glass powder) was applied to prepare internal electrodes. To remove the organic medias in the thick films, the green pellets were sintered at 600 °C for 8 h. The sintering method of MLCCs is the same as above. The obtained MLCCs were washed with alcohol and then coated with silver electrodes. The cross-sectional micro-structure observations can be obtained using a field emission scanning electron microscope (SEM, EMP-800, Tokyo, Japan), the samples were broken and heat treated.

### Electrical properties
The polarization–electric field hysteresis loops (10 Hz) were detected using a ferroelectric test system (Premier II, Radiant Technologies Inc.). The LCR meter (Agilent E4980A, Santa Clara, CA) was adopted to get electric field and temperature dependent dielectric properties (25 ~ 400 °C, 1 kHz-100 kHz) and the in-situ bias dielectric constant-temperature response (−80 ~ 200 °C, 1 kHz-100 kHz, 0 ~ 3000 V). The charge-discharge characteristics were investigated using a specially designed RLC load circuit with high-speed discharge resistance and inductance, the applied resistance is 330 Ω. The room temperature DC breakdown strength measurements were performed using a voltage-withstand testing source (ET2671B, Entai, Nanjing, China). The samples for the above electrical measurements are 2 mm in electrode diameter. The samples for the above electrical measurements are 2 mm in elec-trode diameter, the sample thickness for all electrical tests other than MLCC and complex impedance analysis is 20 ~ 30 μm (Figure S13). Impedance spectroscopy in the frequencies ranging from 150 Hz to 1 MHz within the temperature range of 400 ~ 520 °C can be detected via a LCR meter (Agilent 4284 A, Santa Clara, CA) with a disturbance electric field of 4 V/mm.

### Structural characterization
TEM samples were prepared by mechanical polishing followed by Ar ion-beam milling. EM studies were performed using a spherical aberration-corrected STEM (JEM-ARM200F, JEOL Co. Ltd.) equipped with a cold field-emission gun and a DCOR probe Cs-corrector (CEOS GmbH) operated at 200 kV. The STEM images were obtained using an annular dark-field (ADF) detector with a convergent semi-angle of 20.4

mrad and collection semi-angles of 70 ~ 300 mrad. To obtain accurate lattice constant measurements, ten serial images were acquired with a short dwell time (2 μs/pixel), aligned, and subsequently added to improve the signal-to-noise ratio (SNR) and to minimize the image distortion of HAADF images. Atomic column positions were deter-mined using 2D Gaussian fitting, allowing for the measurements of atomic-scale strain fields and oxygen octahedral rotations with high accuracy. Raman spectra under different electric fields were carried out using a Horiba Lab-Ram HR800 spectrometer with a green laser of 532 nm accompanied by additional heating and voltage-adding devi-ces. All the Raman spectra results were corrected using the Bose-Einstein phonon occupation factor to eliminate the thermal effects on peak intensity. A X-ray diffractometer was used to detect the tem-perature induced phase transition (0.6°/min, 20 ~ 80°, 25 ~ 400 °C). A self-made in-situ electric field XRD sample stage accompanied with a X-ray diffractometer were used to test the phase transition at different applied voltages (0 ~ 2000 V). The samples for in-situ observations possess a comparable thickness as the P-E tests, the applied electrode was specially processed with a semitransparent gold electrode by DC sputtering. A piezo-response force microscope (PFM, Dimension Icon, Bruker, Germany) equipped with a bias voltage-adding device was adopted to detect the surface morphology and the phase angle dis-tribution. The PFM sample was well polished to a roughness of several nanometres and the surface was sputtered with interdigital electrodes. The interdigital electrodes were used to provide a horizontal bias voltage (DC, 0 ~ 1000 V) to drive the AFE-FE phase transition, and the finger gap of interdigital electrodes is about 4 μm, the scanning area is 2 μm * 2 μm, the tip bias is 5 V (AC).

## Data availability
The data that support the findings of this study are available from the corresponding author, see author contributions for specific data sets.

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

## Acknowledgements
We thank Ute Salzberger for assistance with TEM sample preparation, Kersten Hahn, and Tobias Heil for TEM technical support. This work was supported by the Fellowship from the China Postdoctoral Science Foundation (2023M732629). This work is partially funded through the European Union's Horizon 2020 research and innovation programme under grant agreement No. 823717-ESTEEM3, the 19th experimental teaching reformation project of Tongji University (0500104124), and the National Natural Science Foundation of China (52302136).

## Author contributions
G.G.: Conceptualization, Methodology, Investigation, Data curation, Formal analysis, Writing - original draft. J.Q.: Methodology, Investigation, Data curation, Writing - review & editing. C. Shi, and C. Sun: Methodology, Investigation, Software. S.W., and B.S.: Methodology, Investigation. H.W., T.H., and P.A. van Aken: Methodology, Funding acquisition, Writing - review & editing. J.Z.: Funding acquisition, Conceptu-alization, Supervision, Writing - review & editing.

## Funding

## Competing interests
The authors declare no competing interests.
