## [Transparent Peer Review file · Nature Communications]

Structural order differentiation unlocks the energy storage performance of commensurate antiferroelectric ceramics

Corresponding Author: Professor Jiwei Zhai

Version 0:

Reviewer comments:

Reviewer #1

(Remarks to the Author)

Manuscript number: NCOMMS-25-43652-T

Title of the manuscript: "Structural order differentiation unlocks the energy storage performance of commensurate antiferroelectric ceramics" The manuscript presents multilayer ceramic capacitor (MLCC)-type antiferroelectric Pb-based materials and demonstrates high energy storage density. Below are some mandatory comments that the authors must address during the revision of the manuscript.

1. The authors should clarify the rationale for selecting Pb₃O₄ as the lead source instead of the more commonly used PbO. Is there a specific advantage or purpose behind choosing multivalent lead?
2. The authors have specified an electrode diameter of 2 mm; however, it would be beneficial to also provide the sample thickness used for the P-E measurements, as this parameter is essential.
3. The authors should justify their choice of Pb-based materials, especially considering the current research trend focused on replacing lead-based systems with environmentally friendly, lead-free alternatives.
4. Figure 3b should be replotted or enhanced to better visualize the compositional dependence of the properties. The authors should also discuss the behavior of the dielectric maxima. It appears that for Sr = 1%, the dielectric peak lies beyond the measured temperature range, whereas for Sr = 2% and 3%, the peak becomes observable and then shifts to a higher temperature again for Sr = 4%. The authors are requested to provide a detailed explanation for this trend.
5. The authors report promising energy density properties for the Sr = 4% composition, which is the highest concentration studied. However, this raises the question: could even higher Sr content (e.g., 5% or more) yield further improvements? The authors should address whether higher Sr doping was considered or attempted, and if not, explain the reasoning behind limiting the composition range.
6. The authors are encouraged to perform impedance spectroscopy on all samples and analyze their activation energies to gain deeper insight into the electrical behavior of the materials.
7. In Figure 4a, the authors suggest that peak splitting indicates the antiferroelectric (AFE) state, while minimal or no splitting corresponds to the ferroelectric (FE) state. The manuscript would benefit from a more detailed explanation of this interpretation, particularly in relation to structural order and disorder.

Reviewer #2

(Remarks to the Author)

Comments to the Author

The article reports a strategy of structural order differentiation engineering and verified its role in improving the energy storage performance of commensurate modulated antiferroelectric ceramics. The results show that an energy storage density of 23.11 J/cm³, an energy storage efficiency of 85.55%, and a discharge energy density of up to 16.45 J/cm³ were simultaneously achieved. However, there are certain results that would require suitable explanation for better understanding of the facts.

1. What direct experimental evidence supports the distinct roles of Y³⁺ (initiator) and Sr²⁺ (enhancer) in structural order differentiation, and why is 1 mol% Y³⁺ the optimal concentration?
2. What is the change in the oxygen octahedral rotation angle with increasing the Sr content?
3. How does the observed reduction in Pb²⁺ displacement amplitude correlate with the changes in octahedral rotation patterns?

4. Why the electric fields required for antiferroelectric increases with increasing the Sr content?
5. What causes the significant increase in dielectric loss (0.5% → 5%) during the AFE→FE transition?
6. Is there phase coexistence in ceramics with different Sr compositions? How much do the R phase and T phase occupy respectively?
7. How does the degree of disorder in ceramics change with the increasing the Sr content? How does it affect the energy storage performance of ceramics?
8. There are some clerical mistakes in this paper, please check them and correct: such as: "the former is mainly composed of silver niobate or sodium niobate while the latter is is mainly divided into lead zirconate" on page 3, etc.

Version 1:

Reviewer comments:

Reviewer #1

(Remarks to the Author)

The authors have successfully addressed all the comments. The manuscript can be accepted now

Reviewer #2

(Remarks to the Author)

The authors have addressed all my comments. However, the paper still requires some reworking before publication.

1. Based on the XRD refinement results, what are the atomic coordinates and the site occupancies of each element?
2. The authors should quantitatively calculate the theoretical spontaneous polarization of the antiferroelectric S1-S4.
3. Why does applying a horizontal bias electric field drive the AFE→FE phase transition, and why can the induced FE state be stabilized by maintaining this bias?
4. There are still some clerical mistakes in this paper, please check and correct them: such as: "its is..." on page 22, etc.
5. In Fig. S12, numerous voids appear to be present between the layers. How does this affect the performance of MLCCs?

Version 2:

Reviewer comments:

Reviewer #2

(Remarks to the Author)

The authors have answered all the queries well and revised the manuscript accordingly. The revised manuscript is now suitable for publication.

made.

Reviewer #1:

Title of the manuscript: “Structural order differentiation unlocks the energy storage performance of commensurate antiferroelectric ceramics” The manuscript presents multilayer ceramic capacitor (MLCC)-type antiferroelectric Pb-based materials and demonstrates high energy storage density. Below are some mandatory comments that the authors must address during the revision of the manuscript.

1. The authors should clarify the rationale for selecting Pb_3O_4 as the lead source instead of the more commonly used PbO . Is there a specific advantage or purpose behind choosing multivalent lead?

Response:

We thank the reviewer for the valuable comment. When preparing the lead zirconate ceramics, it is feasible to use either PbO or Pb_3O_4 as lead sources. Related reports can be seen in literature, such as “J. Appl. Phys., 2011, 109, 124111” (PbO), “Nat. Commun., 2020, 11, 3809” (Pb_3O_4), and “Adv. Funct. Mater., 2024, 34, 2316674” (Pb_3O_4). In principle, we can consider using one of the two as the lead source. However, in practical operation, we prefer to choose Pb_3O_4 due to the following considerations:

Homogeneity during solid-state reactions: PbO has a lower melting point than Pb_3O_4 , which may lead to premature melting and local inhomogeneities in the mixture during high-temperature processing.

Calcination process monitoring: Unlike PbO , the decomposition of Pb_3O_4 at elevated temperatures is accompanied by a distinct color change from red to yellow, which provides a convenient visual indicator for monitoring phase formation and optimizing calcination conditions.

Chemical environment: PbO is slightly alkaline, whereas Pb_3O_4 is closer to neutral. This difference may significantly influence reaction pathways or product stability in preparation environments with extreme pH conditions.

Furthermore, the use of Pb_3O_4 does not interfere with the crystallization behavior of the resulting ceramic powders, regardless of the specific processing conditions. Taken together, these considerations make Pb_3O_4 a more suitable and reliable lead source for our synthesis approach.

2. The authors have specified an electrode diameter of 2 mm; however, it would be beneficial to also provide the sample thickness used for the *P-E* measurements, as this parameter is essential.

Response:

We thank the reviewer for the suggestion. We have conducted SEM measurements of the cross-section of samples and added the thickness information of the sample in the “Experimental Section” of the revised manuscript (Page 29, lines 16-18; Figure S13). Figure R1 shows the cross-sectional SEM images of PSY1 system for *P-E* tests.

Figure R1. The cross-sectional SEM images of PSY1 system

3. The authors should justify their choice of Pb-based materials, especially considering the current research trend focused on replacing lead-based systems with environmentally friendly, lead-free alternatives.

Response:

We fully understand the concerns of reviewer. The pursuit of lead-free technologies is indeed driven by environmental and health considerations and represents a forward-looking direction. However, the current demands of industrial production and daily applications call for high-performance yet cost-effective

electrostatic capacitor materials, posing significant challenges for the practical implementation of lead-free antiferroelectric ceramics.

While recent advancements have brought the energy storage performance of lead-free antiferroelectric ceramics closer to their lead-based counterparts, key limitations remain. Specifically, materials such as silver niobate and sodium niobate face hurdles including narrow sintering temperature windows, strict atmospheric control requirements, and high raw material costs. In addition, the lead-free systems that demonstrate comparable energy storage properties often involve complex multicomponent compositions (ACS Appl. Mater. Interfaces, 2023, 15, 48354; Nat. Commun., 2025, 16, 2892; J. Mater. Chem. C, 2024, 12, 3962), which complicates their scalability and hinders engineering application.

In contrast, lead zirconate-based antiferroelectric ceramics offer distinct advantages, including compositional simplicity, processability, low material costs, and excellent performance metrics. These features make them strong candidates to meet current market demands for electrostatic capacitors. In the short term, a complete substitution of lead-based capacitors—similar to other lead-containing piezoelectric devices—remains infeasible. This fundamentally justifies our continued focus on lead-based antiferroelectric systems. We also recognize the intrinsic link between lead-free and lead-containing antiferroelectric ceramics in optimizing energy storage performance, such as adjusting the critical temperature to improve energy storage efficiency (Nat. Commun., 2020, 11, 4824).

Therefore, our research aims not only to address current practical needs but also to provide insights into the mechanisms governing energy storage in antiferroelectric materials. We hope that this knowledge will contribute to the future development of cost-effective lead-free alternatives and help bridge the gap between fundamental science and real-world application.

4. Figure 3b should be replotted or enhanced to better visualize the compositional dependence of the properties. The authors should also discuss the behavior of the dielectric maxima. It appears that for Sr = 1%, the dielectric peak lies beyond the measured temperature range, whereas for Sr = 2% and 3%, the peak becomes

observable and then shifts to a higher temperature again for Sr = 4%. The authors are requested to provide a detailed explanation for this trend.

Response:

We understand the concerns of reviewer. After careful consideration, we have decided not to remove the less important curve of dielectric loss from Figure 3b in order to preserve the completeness and clarity of the information presented.

Specifically, for the Sr1 composition, we include the loss curve because the loss peak serves as a valuable complement to the dielectric permittivity peak—particularly in cases where the latter is weak or ambiguous. The definition of the FE_O - “X” phase transition shown in Figure 3b follows the following principle: when the dielectric loss peak is obvious, choose the corresponding temperature; when the dielectric loss peak is not obvious, choose the temperature corresponding to the dielectric anomaly, as Figure S2. The dielectric loss is highly sensitive to structural changes during phase transitions, and it is a commonly used indicator for determining critical temperatures in ferroelectric and antiferroelectric systems. Accordingly, the observed trend of the critical temperature is consistent and justifiable, which is the fundamental reason we chose to retain the current presentation in Figure 3b.

It should also be noted that we believe there should still be dielectric anomaly peaks in higher temperature ranges, thus we defined the “X” state because we are unsure of its symmetry. From the bias dielectric-temperature curve of Sr1, the “X” state may indicate another ferroelectric state rather than a paraelectric state. Therefore, the dielectric maxima mentioned in this research implies the first-order phase transition behavior intrinsic to the induced ferroelectric state.

Meanwhile, we respectfully ask for the reviewer’s understanding regarding the limited temperature range of the dielectric-temperature curves of some compositions. Measuring the dielectric response under bias fields is technically challenging and rarely reported, primarily due to the stringent requirements on sample quality and instrumentation, especially when the bias voltage reaches 2000V or above. Antiferroelectric ceramics are prone to breakdown in coupled fields, which will damage LCR meter. To balance data integrity and equipment safety, we prioritized the collection

of reliable data within the critical temperature range near room temperature, which is most relevant to the energy storage performance targeted in this study. Consequently, the dielectric maxima at higher temperatures were not fully captured in certain compositions.

5. The authors report promising energy density properties for the Sr = 4% composition, which is the highest concentration studied. However, this raises the question: could even higher Sr content (e.g., 5% or more) yield further improvements? The authors should address whether higher Sr doping was considered or attempted, and if not, explain the reasoning behind limiting the composition range.

Response:

We thank the reviewer for the insightful comments. We have conducted additional measurements and provide the P - E loops of PSY1 system with Sr content of 5 mol% and 6 mol%, as shown in Figure R2. It is evident that both compositions exhibit characteristics of polymorphic modulated AFE as described before (<https://doi.org/10.1002/adma.202505731>). The four current peaks within the positive electric field quadrant in the $dP/dt - E$ curves indicate a significant polymorphic modulated AFE. Indeed, if we do not consider the AFE types, more Sr content does lead to a smaller slope of the P - E loops, consistent with our expectations.

In addition, a notable difference is observed between the Sr4 and Sr5 compositions, which are located near the phase boundary between commensurately modulated and polymorphically modulated AFE structures. The former has smaller remanent polarization (less than $1 \mu\text{C}/\text{cm}^2$) and hysteresis ($\sim 58 \text{ kV}/\text{cm}$), while the latter is completely opposite ($\sim 3 \mu\text{C}/\text{cm}^2$ & $\sim 62 \text{ kV}/\text{cm}$). Also, their polarization growth behaviors also differ significantly. The former shows one kind of polarization growth rate of “slope 0”, whereas the latter exhibits the polarization growth rate transition from “slope 1” to “slope 2” with increasing electric field. Regarding energy storage performance, the former has high energy storage efficiency ($\sim 8\%$ higher), and the required electric field is significantly lower than the latter ($\sim 30\%$ lower) when the energy storage density is not significantly enhanced ($\sim 10\%$ higher). These results support our initial decision to limit Sr doping to a maximum of 4 mol% in the main

study, as excessive Sr not only alters the AFE type but also leads to suboptimal performance. Nevertheless, within the Sr doping range of 0–4 mol%, the structural modulation-driven evolution of P – E loop slope is clearly evident.

For completeness, we have included the P – E loops of the 5 mol% and 6 mol% Sr-doped samples in the Revised Supporting Information (Figure S2), though we note that these compositions fall outside the commensurately modulated AFE regime that is the primary focus of this work. We have also added a corresponding explanation in the revised manuscript (Page 13, lines 7–10, highlighted in red font).

Figure R2. The P – E loops of Sr5–Sr6 and the comparison of electrical properties

6. The authors are encouraged to perform impedance spectroscopy on all samples and analyze their activation energies to gain deeper insight into the electrical behavior

of the materials.

Response:

We thank the reviewer for the suggestion. In response, we have conducted high-temperature impedance spectroscopy measurements and provided the corresponding analysis, as shown in the newly added Figure R3. The dataset has also been incorporated into the revised manuscript (Page 25, lines 7-12, highlighted in red font) and the Revised Supporting Information (Figure S11).

High-temperature rather than room-temperature impedance spectroscopy is widely accepted in the characterization of dielectric ceramics, owing to their inherently high resistance at room temperature, such as “J. Am. Ceram. Soc., 2019, 102, 1852-1865”, “J. Appl. Phys., 2019, 125, 084103”, and “Science, 2024, 384, 185-189”. It is evident in the following Nyquist or Cole-Cole plots that the approximate semicircle arc consisting of several frequency-dependent complex impedance responses can be observed. At high frequency regions, there was a weak impedance response process, where the intersection of the high frequency end and the real axis was close to the zero point, indicating that the high frequency part of the impedance spectra came from the bulk response. As the temperature increases from 400 to 520 °C, the impedance semicircles shrink successively, which indicates a thermally activated relaxation process.

Using ZView 2.0, all results are fitted using (RQ)(RQ)(RQ) equivalent circuit instead of the commonly applied (RQ)(RQ) circuit (such as “Mater. Res. Bull., 2024, 173, 112705”, “Adv. Mater., 2025, 37, 2420258”, and “ACS Appl. Mater. Interfaces, 2024, 16, 17787–17796”), each consisting of a parallel ideal resistor R and a constant phase element Q or CPE. As is widely accepted (such as “Science, 2024, 384, 185-189” and “J. Eur. Ceram. Soc., 2018, 38, 4646-4652”), three series of (RQ) units respectively belong to the electrical response of grain boundary, grain, and the polarization of electrodes. The fitting results show small chi-squared function (χ^2) on the order of 10^{-4} , as well as small error of less than 3% for each unit component. The resistance of grain boundaries, grains, and electrodes are labeled here as R_{GB} , R_G , and R_e respectively,

which are usually located in the mid-frequency, high-frequency, and low-frequency regions, and their values decrease in sequence. For bulk conductivity behavior, it is evident that the conductivity can be divided into two levels with Sr doping. The conductivity of Sr1 and Sr2 is higher throughout the entire temperature range, indicating less ideal electrical insulation. Sr3-Sr6 show lower conductivity and thereby their breakdown electric field should be higher than that of Sr1-Sr2. We also separate the contributions of grain boundaries and grains to conductivity, as shown in the middle right figure. Overall, the conductivity of grains is two or three times higher than that of grain boundaries (refer to “J. Alloy. Compd., 2017, 720, 116-125” and “Adv. Energy Mater., 2024, 14, 2304291”). And as the temperature increases, the values of the two become comparable. Thus, it can be inferred that the impedance at room temperature is mainly provided by grain boundaries, which is in line with expectations. For Sr3-Sr5, the contribution of grains on impedance is higher than that of Sr1-Sr2 and Sr6, while the contribution of grain boundaries of Sr5-Sr6 is higher than the rest compositions. This implies that for polymorphic modulated AFE phase, the coexisting of commensurate and incommensurate modulated AFE phases does benefit the bulk insulation. Thus, one can always get higher breakdown electric field in this kind of AFE ceramics, such as “J. Mater. Chem. C, 2021, 9, 12399-12407” and “Appl. Phys. Lett., 2023, 122, 123903”. Using the Arrhenius equation, we can calculate the activation energy (E_a) related to the temperature dependent impedance response. For a pure commensurate modulated AFE matrix, i.e. Sr0-Sr4, the E_a increases with Sr doping content from 0.99 to 1.05 eV. As we all know, in a ABO_3 perovskite structure the A- and B-site cations respectively display E_a of ~ 4 and ~ 12 eV, while that of oxygen vacancies is varied from 0.5 to 2 eV depending on their concentration. Thus, the formation and migration of oxygen vacancies contribute mainly to the ceramic conductivity. We also know that a A-site equivalent substitution mode will not fundamentally produce more oxygen vacancies, and thus the increased E_a should be related to the migration of oxygen vacancies. Therefore, the increase in the E_a from 0.99 to 1.05 eV for the pure commensurate modulated AFE ceramics indicates that the carrier migration necessitates overcoming a higher energy barrier. And in PSY1 system

more Sr doping content leads to a decreased cell volume, which limits the migration of oxygen vacancies and should be the main reason for enhanced E_a (Such as “Ceram. Int., 2021, 47, 26215-26223” and “J. Alloy. Compd., 2023, 947, 169575”). However, for the polymorphic modulated AFE Sr5-Sr6, their E_a decreases with further doping. For Sr5-Sr6, the potential inhomogeneity caused by polymorphic modulated AFE coexistence forms grain boundary and plays a major role in the contribution of conductivity. Therefore, it is not difficult to imagine that for PSY1 system, although a composition induced phase transition occurs with increasing Sr doping, the changes in bulk conductivity are still understandable. The impedance of commensurate modulated AFE phase increases with Sr doping, mainly due to the enhanced E_a at both grain boundaries and grains. When transformed into polymorphic modulated AFE, the competition between grain and grain boundary causes a higher contribution of grain boundaries to the bulk conductivity, which enables the overall insulation to be improved by ignoring the reduced E_a of grains. That is, the factors affecting the breakdown characteristics of the PSY1 system can be summarized as being determined by the E_a , the grain/grain boundary conductivity, and the balance between grain and grain boundaries as a whole. Eventually, phase region end composition Sr4 possesses both lowest grain/grain boundary conductivity as well as highest grain/grain boundary activation energy, it should perform a better breakdown performance than Sr0-Sr3 and is selected as the optimal composition when jointly considering its highly differentiated structural order. Of course, the breakdown electric field of Sr5 and Sr6 can be higher when in the MLCC condition, but this is no longer within the view of pure commensurate modulated AFE phase and will not be further verified here. The corresponding description has been added to the Revised Supporting Information (Figure S11).

Figure R3. High temperature complex impedance spectroscopy of PSY1 system

7. In Figure 4a, the authors suggest that peak splitting indicates the antiferroelectric (AFE) state, while minimal or no splitting corresponds to the ferroelectric (FE) state. The manuscript would benefit from a more detailed explanation of this interpretation, particularly in relation to structural order and disorder.

Response:

We thank for this valuable suggestion. In our revised manuscript, we have added relevant descriptions (Page 19, lines 3-17).

Thank you for your careful evaluation and recognition of our work. We sincerely hope that our responses and related revision address your concerns satisfactorily.

Reviewer #2:

The article reports a strategy of structural order differentiation engineering and verified its role in improving the energy storage performance of commensurate modulated antiferroelectric ceramics. The results show that an energy storage density of 23.11 J/cm³, an energy storage efficiency of 85.55%, and a discharge energy density of up to 16.45 J/cm³ were simultaneously achieved. However, there are certain results that would require suitable explanation for better understanding of the facts.

1. What direct experimental evidence supports the distinct roles of Y³⁺ (initiator) and Sr²⁺ (enhancer) in structural order differentiation, and why is 1 mol% Y³⁺ the optimal concentration?

Response:

We understand the reviewer's concern and appreciate the opportunity to clarify this point. To address the issue, we have supplemented both experimental data and corresponding discussion regarding the effects of Y doping and Sr doping. The following Figure R4 or the updated Figure S1 present the *P–E* loops of the Y-doped PZS system.

Firstly, we need to clarify two points here: firstly, the slope of the *P–E* loops, i.e. the dP/dE , directly displays the intensity of the polarization growth, a steep slope indicates a rapid amplitude polarization switching within a narrow electric field range while a slow slope indicates a slower amplitude polarization switching across a broader electric field range; secondly, the breakdown of BO₆ octahedral distortion in the commensurate modulated AFE phase is a prerequisite for polarization switching (such as “Adv. Mater., 2020, 32, 1907208”). Variations in rotational distortion leads to distinct polarization–electric field responses. Therefore, as shown in the Figure R4, the slope of the *P–E* loop reaches a maximum at the critical electric field. Y doping directly modulates the rate of polarization growth. More notably, at higher Y doping levels, the *P–E* loops become less square-like and more slanted, with a clear decrease in slope near the AFE–FE transition. Correspondingly, the $dP/dE–E$ curves show a reduced peak magnitude and a broader peak width. The latter indicates that polarization switching occurs over a wider range of electric fields, revealing a more diffuse polarization

response. This diffuseness is beneficial for energy storage applications, as it helps reduce energy loss and improve cycling stability. However, excessive Y doping severely suppresses the net polarization of the PZS matrix. Therefore, a 1 mol% Y doping level is found to be optimal, achieving a balance of high polarization, low hysteresis, and moderate slope, all of which are essential for enhanced energy storage performance. The relevant description has been added to the Revised Manuscript on Page 13, lines 3-7.

Building upon the effect of Y doping, Sr doping further suppresses the polarization growth rate (dP/dE). From the HAADF results, the A-site cation displacement or polarization mode remains four-fold periodic modulation, while the BO_6 octahedra distribution breaks its original “++--” ordered arrangement with Sr doping. The “+” and “-” here represent the left and right rotations of an octahedra, respectively. For Sr4, there are some disordered octahedral sequences such as “+++” or “---+” interspersed in the basic “++--” arrangement. The statistical results show that when doped with 1 mol% Sr, the rotation mode of the octahedra exhibits a roughly symmetrical bimodal shape distribution of left- and right-rotated. When doped with 4 mol% Sr, this bimodal distribution becomes unimodal, and the rotation mode of the octahedra is mainly near zero-degree rotation, accompanied by a certain amount of left- and right-rotated modes with different rotation angles. This is defined as the differentiation process of structural order, where the set of rotation angles is greatly expanded.

The results of joint P - E loops and in-situ structure evolution, such as the broadening of dielectric anomaly peaks in bias dielectric-temperature response or the sensitivity of domain structure evolution to electric fields in PFM testing, indicate that the structural order differentiation directly prolongs the polarization switching process, resulting in increasingly slanted P - E loops. In summary, the effect of Sr or Y doping is similar, both playing a role in regulating octahedral rotational distortion, which directly promotes the diversification of structural order and thus affects the process of AFE-FE phase transition, making the energy storage performance of commensurate modulated AFE no longer limited by the critical electric field. While their roles are complementary, the effects of the initiator dopant Y and the enhancer dopant Sr are not fundamentally

distinct. Y doping perturbs the highly square-like P - E response of the pristine PZS matrix, while Sr doping builds upon this perturbation and amplifies the structural modulation effect. It should be noted that Y is a donor doping, and excessive Y incorporation can introduce charged point defects, which degrade polarization and breakdown strength. In contrast, the reason for choosing the equivalent ion Sr as an enhancer is that it does not construct additional defects but at least maintains the polarization and breakdown characteristics. Therefore, Sr can be used to adjust the structural order without changing the polarization order.

Figure R4. The electrical properties of $\text{Pb}_{1-1.5x}\text{Y}_x\text{Zr}_{0.6}\text{Sn}_{0.4}\text{O}_3$ system

2. What is the change in the oxygen octahedral rotation angle with increasing the Sr content?

Response:

As illustrated in Figure R5, the real lattice structure of lead zirconate features A-site cation displacements (blue spheres) oriented along the $[110]_p$ and $[\bar{1}\bar{1}0]_p$ pseudocubic directions, while the BO_6 octahedra exhibit antiphase tilting, i.e., left- and right-handed rotations. In a simplified scenario where there are only two kinds of octahedra rotation with rotation angle of $\pm\theta^0$, the AFE-FE phase transition undergoes a simple process of AFE/AFD-FD-FE. And thereby the P - E loops display square features

approximately perpendicular to the electric field coordinate axis. This indicates that polarization grows abruptly once the critical electric field is reached. The breakdown of BO_6 octahedral distortion in the commensurate modulated AFE phase is a prerequisite for polarization switching.

However, when the rotational distortion of the BO_6 octahedra becomes more complex—with a broader distribution of rotation angles (θ^+ , θ^0 , θ^- ,)—the disruption of octahedral tilting becomes increasingly intricate. In this case, multiple AFD-FD phase transitions appear in different lattices with different octahedral distortion, resulting in increasingly slanted P - E loops.

Thus, the change in the oxygen octahedral rotation angle with increasing the Sr content can be simply summarized as the increase (θ^+) and decrease (θ^-) in the rotation angle of octahedra induced by cell volume effect. The Coulomb effect of A-site substitution leads to the change in the center of gravity of cations, and thereby causes the re-distribution of the oxygen anions and facilitates the rotation and re-rotation of octahedra.

Figure R5. The schematic diagram of Sr doping effect on polarization growth

3. How does the observed reduction in Pb^{2+} displacement amplitude correlate with the changes in octahedral rotation patterns?

Response:

We thank the reviewer for the comment. Based on the current experimental data, we suggest that it is less appropriate to correlate them together. The presence of A-site ion substitution can reduce the concentration of lead cations to some extent. In this case, the average cation displacement is inevitably influenced by A-site ions with small displacement. In contrast, an octahedra can be seen as a whole, the effect of A-site cation on the octahedra is always reflected in the obstruction of movement space by reducing the unit cell volume. It is difficult to link octahedra that undergo larger or smaller angle rotation with polarization within the lattice, as lattice distortions in adjacent lattices transmit stress to the local lattice. That is, the changes in polarization displacement and octahedral rotation distortion may not necessarily manifest in the same lattice at the same time. Taking all factors into consideration, we may not be able to fully match these two at present. Kindly looking for your understanding.

4. Why the electric fields required for antiferroelectric increases with increasing the Sr content?

Response:

We thank the reviewer for the insightful comment. As far as we know, the phase stability of antiferroelectric is related to the unit cell volume, such as “J. Am. Ceram. Soc., 2021, 104, 3775-3810”, “J. Am. Ceram. Soc., 2011, 94, 4091-4107”, and “J. Am. Ceram. Soc., 2022, 105, 6765-6774”. Specifically, the volume of the unit cell affects the rotation angle of the octahedra, that is, the stability of the structural order. In the case of a larger unit cell volume, octahedra are more prone to rotation in larger spaces, leading to distortion release and structural instability and vice versa. Considering that the instability of structural order is a prerequisite for the antiferroelectric-ferroelectric phase transition, a larger unit cell volume means that the difficulty of the antiferroelectric-ferroelectric phase transition is reduced, and the critical field is therefore lowered. Sr cation is smaller than Pb cation, the unit cell volume is reduced when Sr substitutes Pb. This is also confirmed by the XRD results, where the cell

volume of the average structure decreases with Sr doping. Thus, the electric field required for antiferroelectric increases with Sr doping.

5. What causes the significant increase in dielectric loss (0.5% → 5%) during the AFE→FE transition?

Response:

We thank the reviewer for the insightful comment. This is indeed a very interesting topic. However so far, a very definite conclusion on this issue remains elusive at present. We provide the following response from the basic structure of antiferroelectric.

Antiferroelectricity is characterized by short-range ordered antiparallel dipoles, in contrast to the long-range parallel dipole alignment found in ferroelectric (FE) materials. The unique polarization configuration patterns result in macroscopically near-zero polarization. This means that in the cases of zero or small electric field, AFE materials do not retain excessive polarized charges, reducing energy loss caused by remanent polarization. However, in ferroelectric materials, dipoles are arranged in parallel, and the macroscopically remanent polarization cannot be ignored. Under a limited electric field, the re-orientation or rearrangement of dipoles in AFE is much simpler and more ordered than that in FE materials. That is, a large energy barrier needs to be overcome to change the orientation of dipoles in FE, this results in more energy loss. In addition, AFE materials undergo a reversibly AFE-FE-AFE phase transition under an electric field. This reversible phase transition process enables energy to be efficiently converted, which reduces the energy loss. While in many cases the phase transition in ferroelectric materials may not be completely reversible, resulting in higher energy loss. At the same time, the transformation from AFE to FE requires additional electrical energy, and from an energy perspective antiferroelectric states are more stable. Therefore, for ferroelectric lattices, small energy inputs such as dielectric measurements may cause changes in lattice constants and the movement of lattice defects, which consume most of the energy and results in additional energy losses such as lattice vibration losses. The relative stability of the lattice structure of AFE helps to reduce such losses. Thus, we can distinguish between ferroelectric and antiferroelectric states by changes in dielectric loss. That is, when an AFE-FE phase transition occurs, a significant increase

in dielectric loss means a transition from the antiferroelectric state to the ferroelectric state.

6. Is there phase coexistence in ceramics with different Sr compositions? How much do the R phase and T phase occupy respectively?

Response:

We acknowledge the comment of the reviewer. Based on existing research, we propose that phase coexistence is likely present in the PSY1 system. Nevertheless, this phase coexistence is difficult to determine through XRD and Raman spectroscopy. At the macro scale, i.e. the average structure, we consider the phase structure to be single. However, at the microscale, if a lattice with different octahedral rotation modes is defined as a phase structure, we consider the coexistence of phase structures to be fluctuating at the microscale. Therefore, the PSY1 system is a single commensurate modulated AFE phase that is microscopically non-uniform but macroscopically uniform.

As for the ratio of R and O phases, we attempted to refine it but did not obtain reliable results due to the narrow 2θ range and the less ideal diffraction intensity limited by the electrode and silicone oil. Here we provide another method to roughly determine the relative content of two phases by judging the intensity ratio of (200)/(002) indexed diffraction. As described in the red font on Page 19 of the revised manuscript, ferroelectric R and O phases have a distinct (200)/(002) intensity ratio. The ferroelectric state of Sr1 has an intensity ratio greater than 8:1, which means that the ratio of ferroelectric R to ferroelectric O exceeds 3:1. In Sr4, the ratio of ferroelectric R to ferroelectric O should be slightly less than 2:1. This is consistent with the results of the bias dielectric-temperature response curves, where the diffuse dielectric anomaly peak mean a diffusion of ferroelectric O phase into the ferroelectric R phase. Of course, this kind of phase coexistence is still a reflection of the microscale lattice fluctuations by the large-scale average structure, that is a statistical result, and should not be seen as coexistence in the usual sense. These two ferroelectric phases diffuse into each other at the lattice scale, as the distribution of octahedra with different rotation modes is uncertain (Figure R5 and Figure 2). It is better to consider this coexistence as an

interweaving of phases at the lattice scale rather than treating them separately.

7. How does the degree of disorder in ceramics change with the increasing the Sr content? How does it affect the energy storage performance of ceramics?

Response:

As mentioned above, there are some disordered octahedral sequences such as “++-+” or “---+” interspersed in the basic “+++” arrangement. Thus, the degree of disorder in ceramics gets enhanced with the Sr doping content if comparing Sr1 and Sr4 merely. The HAADF images of the rest compositions are lacked, but we believe that the degree of disorder gradually increases with Sr doping as the in-situ phase transition results and the electrical properties echo each other very well. The strengthened disorder leads to a multi-stage AFD-FD transition as mentioned above, this delays the AFE-FE phase transition and prolongs the polarization growth process. And meanwhile, the maximum polarization of Sr1-Sr4 is comparable. Therefore, the energy storage performance of PSY1 system gets improved with the strengthened disorder. However, further increasing the degree of disorder may lead to instability of modulation structures, such as Sr5 and Sr6, and the changes in energy storage performance will be unpredictable.

8. There are some clerical mistakes in this paper, please check them and correct: such as: “the former is mainly composed of silver niobate or sodium niobate while the latter is is mainly divided into lead zirconate” on page 3, etc.

Response:

We acknowledge the reviewer’s careful review and appreciate the attention to detail. We have made corrections to some inappropriate expressions or errors that appeared in the article.

Thank you for your careful evaluation thoughtful on our research. We sincerely hope that the modifications and the responses will satisfy you. May everything go well and happily with you.

Reviewer #1:

The authors have successfully addressed all the comments. The manuscript can be accepted now.

Response:

Thank you for your recognition and support.

Reviewer #2:

The authors have addressed all my comments. However, the paper still requires some reworking before publication.

Response: We sincerely thank the reviewer for the careful evaluation and rigorous attitude in reviewing our work. The constructive comments have prompted us to reflect more deeply on our study and have led to significant improvements in the clarity and quality of the manuscript.

1. Based on the XRD refinement results, what are the atomic coordinates and the site occupancies of each element?

Response:

We understand the reviewer's concern and appreciate the opportunity to clarify this. In response, we have provided the refined positions of occupying ions based on the XRD refinement. The results confirm that the ion occupancies are consistent with the expected stoichiometry, thereby meeting the requirements at this level.

However, we would like to stress the inherent limitations of XRD refinements for determining precise atomic positions in this system. Specifically, the use of Cu K α radiation is not suitable for accurately probing the occupancy of light elements such as oxygen, and neutron diffraction, better suited for this purpose, was not available in the present study. For this reason, we did not attempt to refine oxygen positions, but instead adjusted only the heavy-atom positions to match standard structures such as pure lead zirconate. The heavy atoms define the primary lattice framework even without considering internal octahedral distortions.

The refinement results show that both the main and superlattice diffractions remain consistent with lead zirconate, and the observed changes in cell parameters can be attributed to ion substitution. Thus, the phase determination is reliable. Nevertheless, the absence of detectable octahedral rotational distortions in the refinement contradicts our direct experimental measurements, further demonstrating the limitations of XRD in this context.

Concerning the dopants, Sr and Y were placed at the 4g position in the refinement, though placement at the 4h position is also possible when ionic radii and coordination

environments are considered. Due to their very low concentrations, it is not possible to unambiguously assign their occupancy. From a symmetry perspective, it is unlikely that such dilute dopants could stabilize a C_5 -symmetric structure. Therefore, under these conditions, the principal significance of the XRD refinement lies in establishing the overall spatial symmetry and estimating unit cell parameters, rather than providing precise atomic positions.

We also note that in recent high-impact publications in the field of energy-storage ceramics (e.g., *Science* 2024, 384, 185–189; *Nat. Commun.* 2025, 16, 1633; *Adv. Mater.* 2024, 36, 2410088), detailed atomic positions derived from XRD refinements are generally not reported, since excessive interpretation of less precise parameters can be misleading. In our work, XRD refinements are presented as supportive evidence, while the decisive structural insights are provided by atomic-scale characterization, which offers higher precision and direct information on ordering.

We hope this explanation addresses the reviewer's concern and clarifies the reasoning behind our approach.

	Atom	Ox.	Wyckoff position	Site symmetry	x/a	y/b	z/c	occupancy
Sr=1%	Pb1	2	4g	..m	0.6967	0.124856	0	0.9682
	Pb2	2	4h	..m	0.6967	0.124856	0.5	0.9863
	ZR1	4	8i	1	0.238721	0.122859	0.247559	0.6089
	O1	-2	4g	..m	0.271	0.156	0	1
	O2	-2	4h	..m	0.288	0.096	0.5	1
	O3	-2	8i	1	0.032	0.2608	0.283	1
	O4	-2	4f	..2	0	0.5	0.206	1
	O5	-2	4e	..2	0	0	0.248	1
	Sn1	4	8i	1	0.238721	0.122859	0.247559	0.3911
	Sr1	2	4g	..m	0.6967	0.124856	0	0.0098
	Y1	3	4g	..m	0.6967	0.124856	0	0.0097
Sr=2%	Pb1	2	4g	..m	0.6996	0.125617	0	0.9605
	Pb2	2	4h	..m	0.6996	0.125617	0.5	0.9795
	ZR1	4	8i	1	0.240012	0.120946	0.247451	0.5915
	O1	-2	4g	..m	0.271	0.156	0	1
	O2	-2	4h	..m	0.288	0.096	0.5	1
	O3	-2	8i	1	0.032	0.2608	0.283	1
	O4	-2	4f	..2	0	0.5	0.206	1
	O5	-2	4e	..2	0	0	0.248	1
	Sn1	4	8i	1	0.240012	0.120946	0.247451	0.4085
	Sr1	2	4g	..m	0.6996	0.125617	0	0.0209
	Y1	3	4g	..m	0.6996	0.125617	0	0.0098
Sr=3%	Pb1	2	4g	..m	0.7013	0.123671	0	0.954
	Pb2	2	4h	..m	0.7013	0.123671	0.5	0.9852
	ZR1	4	8i	1	0.238532	0.121421	0.246896	0.5975
	O1	-2	4g	..m	0.271	0.156	0	1
	O2	-2	4h	..m	0.288	0.096	0.5	1
	O3	-2	8i	1	0.032	0.2608	0.283	1
	O4	-2	4f	..2	0	0.5	0.206	1
	O5	-2	4e	..2	0	0	0.248	1
	Sn1	4	8i	1	0.238532	0.121421	0.246896	0.4025
	Sr1	2	4g	..m	0.7013	0.123671	0	0.0299
	Y1	3	4g	..m	0.7013	0.123671	0	0.0098
Sr=4%	Pb1	2	4g	..m	0.7042	0.125782	0	0.9427
	Pb2	2	4h	..m	0.7042	0.125782	0.5	0.9831
	ZR1	4	8i	1	0.239103	0.123068	0.253258	0.5962
	O1	-2	4g	..m	0.271	0.156	0	1
	O2	-2	4h	..m	0.288	0.096	0.5	1
	O3	-2	8i	1	0.032	0.2608	0.283	1
	O4	-2	4f	..2	0	0.5	0.206	1
	O5	-2	4e	..2	0	0	0.248	1
	Sn1	4	8i	1	0.239103	0.123068	0.253258	0.4038
	Sr1	2	4g	..m	0.7042	0.125782	0	0.0391
	Y1	3	4g	..m	0.7042	0.125782	0	0.0107

Figure R1. The ionic occupancy of $\text{Pb}_{0.985-x}\text{Sr}_x\text{Y}_{0.01}\text{Zr}_{0.6}\text{Sn}_{0.4}\text{O}_3$ system

2. The authors should quantitatively calculate the theoretical spontaneous polarization of the antiferroelectric S1-S4.

Response:

We thank the reviewer for this valuable comment. In response, we have carried out a manual estimation of the spontaneous polarization in the antiferroelectric samples S1–S4 based on the displacements of A-site and B-site cations obtained from XRD refinements. Specifically, we considered Pb ions at the 4g position as the dominant contributors to the A-site polarization, while Sr and Y were neglected due to their very

low concentrations. From the refinement results, the Pb displacements in S1–S4 were calculated as 31.04, 29.34, 28.35, and 26.66 pm, corresponding to local polarizations of 28.07, 26.55, 25.65, and 24.13 $\mu\text{C}/\text{cm}^2$, respectively, along the pseudo-cubic [110] direction. In addition, Zr ions in PbZrO_3 also contribute partial polarization along this direction. Assuming Sn behaves similarly to Zr, our calculations yield values of 8.90, 7.88, 9.05, and 8.60 $\mu\text{C}/\text{cm}^2$ for S1–S4, respectively, which is consistent with previous reports that Zr displacement accounts for about one-third of the Pb contribution. The total cationic polarization along [110] is estimated as 36.97, 34.43, 34.71, and 32.74 $\mu\text{C}/\text{cm}^2$ for S1–S4, respectively.

These values are broadly consistent with polarization vector distribution maps, which indicate maximum spontaneous polarizations of 29.31 $\mu\text{C}/\text{cm}^2$ (S1) and 28.61 $\mu\text{C}/\text{cm}^2$ (S4), with average A-site-related polarizations of ~ 19.02 $\mu\text{C}/\text{cm}^2$ (S1) and ~ 16.77 $\mu\text{C}/\text{cm}^2$ (S4). The small discrepancies between the two approaches may originate from slight zone-axis tilting during measurements or from the limited accuracy of XRD refinements using Cu $K\alpha$ radiation.

Nevertheless, we would like to emphasize the limitations of this analysis. Reliable polarization calculations require both Born effective charges and precise atomic coordinates. While Born effective charges are available in the literature (e.g., Phys. Rev. B 1999, 60, 836–843), the oxygen anion positions refined from Cu $K\alpha$ XRD remain unchanged with doping and cannot be determined accurately without more refined structural analysis methods. This uncertainty limits the reliability of polarization estimates derived solely from XRD refinements. In fact, for reference, in pure PbZrO_3 , calculated cationic polarizations are typically ~ 40 $\mu\text{C}/\text{cm}^2$ (e.g., Nat. Commun. 2020, 11, 3809), whereas experimentally measured values are significantly higher (e.g., J. Eur. Ceram. Soc. 2019, 39, 4761–4769).

Therefore, more refined structural analysis may not fully meet our further expectations for data validity and completeness in providing more detailed evidence, and in practical operation we also lack the corresponding experimental conditions and experience accumulation. Our calculations here should be regarded only as approximate and illustrative, and we believe that the currently presented results can

explain the main issues. We therefore may suggest relying primarily on experimental results for reliable evaluation of spontaneous polarization in these ceramics.

3. Why does applying a horizontal bias electric field drive the AFE–FE phase transition, and why can the induced FE state be stabilized by maintaining this bias?

Response:

We understand the concern of the reviewer. The antiferroelectric state is the ground state, in the lowest energy state. The electric field changes the phase structure, manifested as changes in ions displacement, causing the polarization and structural order of the antiferroelectric state to become unstable. The antiferroelectric ground state is no longer stable and transforms into a ferroelectric state. The phase transition from antiferroelectric to ferroelectric state is not spontaneous, whereas the transition from ferroelectric state to antiferroelectric state is spontaneous. Therefore, if studying the characteristics of ferroelectric states, it is necessary to maintain the electric field. Consider a simplest parallel plate capacitor, where the direction of electric field is from top to bottom. However, during PFM testing, air breakdown may occur directly between the probe and the electrode, which will damage the equipment. Therefore, it is necessary to avoid possible “contact” between the probe and the electrode while applying an electric field. So, we stabilize the ferroelectric state by applying a horizontal electric field, and scan the phase angle distribution of the sample surface within the electrode gap. In addition, because ceramics are polycrystalline, the anisotropy of the lattice is averaged on a large scale, the electric fields applied in the horizontal and vertical directions are comparable, they can both drive the antiferroelectric-ferroelectric phase transition.

4. There are still some clerical mistakes in this paper, please check and correct them: such as: "its is..." on page 22, etc.

Response:

We thank the reviewer for pointing out this. In this revised version, we have reviewed the manuscript again, the corresponding errors or the less appropriate expressions have been corrected, as indicated in the red font in the revised manuscript.

5. In Fig. S12, numerous voids appear to be present between the layers. How does

this affect the performance of MLCCs?

Response:

We thank the reviewer for the suggestion. The voids only appear in the electrode layer, and the dielectric layer is rather dense. The actual test results show good conductivity, and the effective electrode area conforms to the design. The electrode slurry we selected is Pt-75, which contains active zirconia powder. The particle size of zirconia powder ranges from tens to hundreds of nanometers, mainly used to match the sintering temperature. At high temperatures, both Pt and ZrO₂ crystallize, together forming the microstructure of the electrode layer. Thus, we believe that these voids originate from the sample post-processing before cross-sectional observation. The voids can be considered as the detachment of Pt or ZrO₂ grains during ceramic fracture or polishing processes. For the sintered MLCCs, the electrodes are essentially dense and continuous, and the commercialized electrode paste meets our expectations. And the voids caused by the post-processing should not affect the performance of the untreated MLCC samples.

Thank you for your careful evaluation and thoughtful suggestions on our research. We sincerely hope that the modifications and the responses will satisfy you.